# Comprehensive mapping of adaptation of the avian influenza polymerase protein PB2 to humans

YQ Shirleen Soh[1,2], Louise H Moncla[2,3], Rachel Eguia[1], Trevor Bedford[2,3], Jesse D Bloom[1,2,4]*

[1]Basic Sciences Division, Fred Hutchinson Cancer Research Center, Seattle, United States; [2]Computational Biology Program, Fred Hutchinson Cancer Research Center, Seattle, United States; [3]Vaccine and Infectious Disease Division, Fred Hutchinson Cancer Research Center, Seattle, United States; [4]Howard Hughes Medical Institute, Seattle, United States

**Abstract** Viruses like influenza are infamous for their ability to adapt to new hosts. Retrospective studies of natural zoonoses and passaging in the lab have identified a modest number of host-adaptive mutations. However, it is unclear if these mutations represent all ways that influenza can adapt to a new host. Here we take a prospective approach to this question by completely mapping amino-acid mutations to the avian influenza virus polymerase protein PB2 that enhance growth in human cells. We identify numerous previously uncharacterized human-adaptive mutations. These mutations cluster on PB2's surface, highlighting potential interfaces with host factors. Some previously uncharacterized adaptive mutations occur in avian-to-human transmission of H7N9 influenza, showing their importance for natural virus evolution. But other adaptive mutations do not occur in nature because they are inaccessible via single-nucleotide mutations. Overall, our work shows how selection at key molecular surfaces combines with evolutionary accessibility to shape viral host adaptation.
DOI: https://doi.org/10.7554/eLife.45079.001

*For correspondence: jbloom@fredhutch.org

**Competing interests:** The authors declare that no competing interests exist.

## Introduction

Viruses are exquisitely adapted to interact with host-cell machinery to facilitate their replication. Despite significant differences in this machinery across host species, some viruses like influenza can evolve to infect divergent hosts (*Parrish et al., 2008*; *Webster et al., 1992*). Such zoonotic transmissions can have severe public health consequences: transmission of influenza virus from birds or pigs to humans has resulted in four pandemics over the last century (*Taubenberger and Kash, 2010*). These pandemics require the virus to adapt to the new host (*Long et al., 2019*). Delineating how viruses adapt to new hosts will aid in our ability to understand what determines if a chance zoonotic infection evolves into a human pandemic.

One critical determinant of influenza host range is the viral polymerase (*Long et al., 2019*), which transcribes and replicates the viral genome (*Eisfeld et al., 2015*; *te Velthuis and Fodor, 2016*). Avian influenza polymerases perform poorly in mammalian cells (*Cauldwell et al., 2014*; *Mänz et al., 2012*; *Massin et al., 2001*; *Naffakh et al., 2000*). This host range restriction likely arises from the need for the viral polymerase to interact with host proteins such as importin-α (*Resa-Infante and Gabriel, 2013*) and ANP32A (*Long et al., 2016*), which differ between avian and mammalian hosts. However, it remains unclear exactly how the molecular interfaces between the polymerase and these host proteins are altered during adaptation to humans (*Long et al., 2019*).

**eLife digest** Viruses copy themselves by hijacking the cells of an infected host, but this comes with some limitations. Cells from different species have different molecular machinery and so viruses often have to specialize to a narrow group of species. This specialization consists largely of fine-tuning the way that viral proteins interact with host proteins.

For instance, in bird flu viruses, a protein known as PB2 does not interact well with the machinery in human cells. Because PB2 proteins form part of the viral polymerase (the structure that copies the viral genome), this prevents bird flu viruses from replicating efficiently in humans. Sometimes however, changes in the PB2 protein allow bird flu viruses to better replicate in humans, potentially leading to deadly flu pandemics.

To understand exactly how this happens, researchers have previously used two approaches: examining the changes that have happened in past flu viruses, and monitoring the evolution of bird flu viruses grown in human cells in the lab. However, these approaches can only look at a small number of the many possible genetic changes to the virus. This makes it hard to anticipate the new ways that flu might adapt to human cells in the future.

To overcome this problem, Soh et al. systematically created all of the single changes to the bird flu PB2, altering every element of the protein sequence one-by-one. They then tested which of the changes to PB2 helped the virus grow better in human cells. The modifications that made the viruses thrive were on the surface of the protein, suggesting that they might improve interaction with the cell machinery of the host. Some changes have been found in bird flu viruses that have recently jumped into humans in nature, although fortunately none of these viruses have yet spread widely to cause a pandemic.

Many factors affect the evolution of viruses, and their ability to infect new species. Understanding which changes in proteins help these microbes adapt to new hosts is an important element that scientists could consider to assess future risks of pandemics.

DOI: https://doi.org/10.7554/eLife.45079.002

Studies of natural zoonoses and experimental passaging of viruses in the lab have identified a number of mutations that adapt avian influenza polymerases to mammalian hosts (*Bussey et al., 2010*; *Cauldwell et al., 2014*; *Cauldwell et al., 2013*; *Chen et al., 2006*; *Finkelstein et al., 2007*; *Gabriel et al., 2005*; *Hu et al., 2017*; *Hu et al., 2014*; *Kim et al., 2010*; *Mänz et al., 2016*; *Mehle and Doudna, 2009*; *Miotto et al., 2008*; *Mok et al., 2011*; *Naffakh et al., 2000*; *Reperant et al., 2012*; *Taft et al., 2015*; *Tamuri et al., 2009*; *Yamada et al., 2010*; *Zhou et al., 2011*). The best known of these mutations is E627K in the PB2 subunit of the polymerase (*Subbarao et al., 1993*). This mutation alone significantly improves avian influenza polymerase activity in mammalian cells (*Long et al., 2013*; *Massin et al., 2001*; *Mehle and Doudna, 2008*; *Naffakh et al., 2000*), and was considered a key step in adaptation to humans (*Taubenberger et al., 2005*). But surprisingly, the recent 2009 H1N1 pandemic lineage lacks the E627K mutation. Instead, it has acquired mutations to PB2 at sites 590 and 591 that similarly confer improved polymerase activity (*Mehle and Doudna, 2009*; *Yamada et al., 2010*). This fact underscores the possibility that natural evolution has explored only a small fraction of the possible host-adaptation mutations. Examining only the currently available instances of adaptation in nature or the lab may therefore overlook additional mechanisms of adaptation and evolutionary paths to future zoonoses.

Here, we map all single amino-acid mutations to an avian influenza PB2 protein that enhance growth in human cells versus avian cells. We do so by leveraging deep mutational scanning (*Boucher et al., 2014*; *Fowler and Fields, 2014*), which previously has only been used to measure the functional effects of mutations to several influenza proteins in mammalian cells (*Ashenberg et al., 2017*; *Bloom, 2014*; *Doud and Bloom, 2016*; *Du et al., 2018*; *Jiang et al., 2016*; *Lee et al., 2018*; *Wu et al., 2014a*; *Wu et al., 2014b*). We show that comparative deep mutational scanning in human versus avian cells identifies numerous human-adaptive mutations that have never before been described. These mutations cluster on the surface of the PB2 protein, highlighting potential interfaces with host factors. Some of these mutations are enriched in avian-human

transmission of H7N9 influenza, demonstrating the utility of our experiments for anticipating PB2's adaptation in nature. The human-adaptive mutations that have not been observed in nature are often inaccessible by single-nucleotide mutations. Overall, our complete map of human-adaptive mutations sheds light on how species-specific selection and evolutionary accessibility shape influenza virus's evolution to new hosts.

## Results

### Deep mutational scanning of an avian influenza PB2

To identify host-adaptation mutations in PB2, we used deep mutational scanning to measure the effects of all amino-acid mutations to this protein in both human and avian cells. We performed these experiments using the PB2 from an avian influenza strain, A/Green-winged Teal/Ohio/175/1986 (also previously referred to as S009) (*Jagger et al., 2010*; *Mehle and Doudna, 2009*). The PB2 from this strain is representative of avian influenza PB2s, most of which are highly similar (average pairwise amino-acid identity of 98.7%) (*Figure 1—figure supplement 1*). We mutagenized all codons in PB2 to create three replicate mutant plasmid libraries with an average of 1.4 codon substitutions per clone (*Figure 1A*, *Figure 1—figure supplement 2A–F*). Since there are 759 residues in PB2, there are 759 × 19 = 14,421 amino acid mutations, virtually all of which are represented in our libraries (*Figure 1—figure supplement 2G*).

We generated a mutant virus library from each of the triplicate plasmid mutant libraries using a helper-virus approach, which reduces bottlenecks during generation of complex viral libraries (*Doud and Bloom, 2016*) (*Figure 1A*). For biosafety reasons, we rescued reassortant virus using polymerase (PB2, PB1, PA) and nucleoprotein (NP) genes from the avian influenza strain and the remaining viral genes (HA, NA, M, NS) from the lab-adapted A/WSN/1933(H1N1) mammalian influenza strain. We wanted to minimize selection for host-adaptive mutations during the initial library generation. Therefore, we generated the libraries in human HEK293T cells with a co-transfected protein-expression plasmid encoding the human-adapted PB2-E627K protein variant, so that all cells had a PB2 protein that could complement poorly functioning library variants.

To select for functional PB2 variants in human versus avian cells, we passaged each replicate mutant virus library at low MOI in the A549 human lung epithelial carcinoma line and CCL-141 duck embryonic fibroblasts (*Figure 1A*, *Figure 1—figure supplement 2A*). To quantify the functional selection on each mutation during viral growth, we deep sequenced the initial plasmid mutant libraries and the passaged mutant viruses to measure the frequency of mutations before and after selection (*Figure 1B*, *Figure 1—figure supplement 3*). All experiments were also performed in parallel on virus carrying wild-type PB2 as a control to quantify the rate of errors arising during sequencing, library preparation, and viral replication (*Figure 1—figure supplement 2H*).

To assess the efficacy of selection without the complication of errors arising from sequencing and passaging, we examined the post-selection frequency of stop and nonsynonymous mutations accessible by >1 nucleotide substitution. Stop and nonsynonymous mutations fell to 2–7% and 26–35% of their initial frequencies respectively (*Figure 1—figure supplement 2H*). In contrast, synonymous mutations remained at 68–87% of their initial frequency. Therefore, the experiments effectively selected for functional PB2 mutants.

We quantified selection at the amino-acid level in terms of the 'preference' of each site in the protein for each amino acid (*Figure 1B*) (*Bloom, 2015*). The preference for an amino acid is proportional to its enrichment during functional selection. We assessed the reproducibility of our experiments across biological replicates by examining the correlations of preferences for all 14,421 amino acid (*Figure 1—figure supplement 2I*). Biological replicates passaged in each cell type were well correlated (Pearson's *R* in human cells was 0.74 to 0.79; Pearson's *R* in avian cells was 0.76 to 0.79), and were generally better correlated within cell types than between cell types (Pearson's *R* between cell types was 0.67 to 0.78). For downstream analyses, we rescaled our preferences to match the stringency of selection in nature (see Materials and methods, *Supplementary file 4*, *Figure 2—source data 1*).

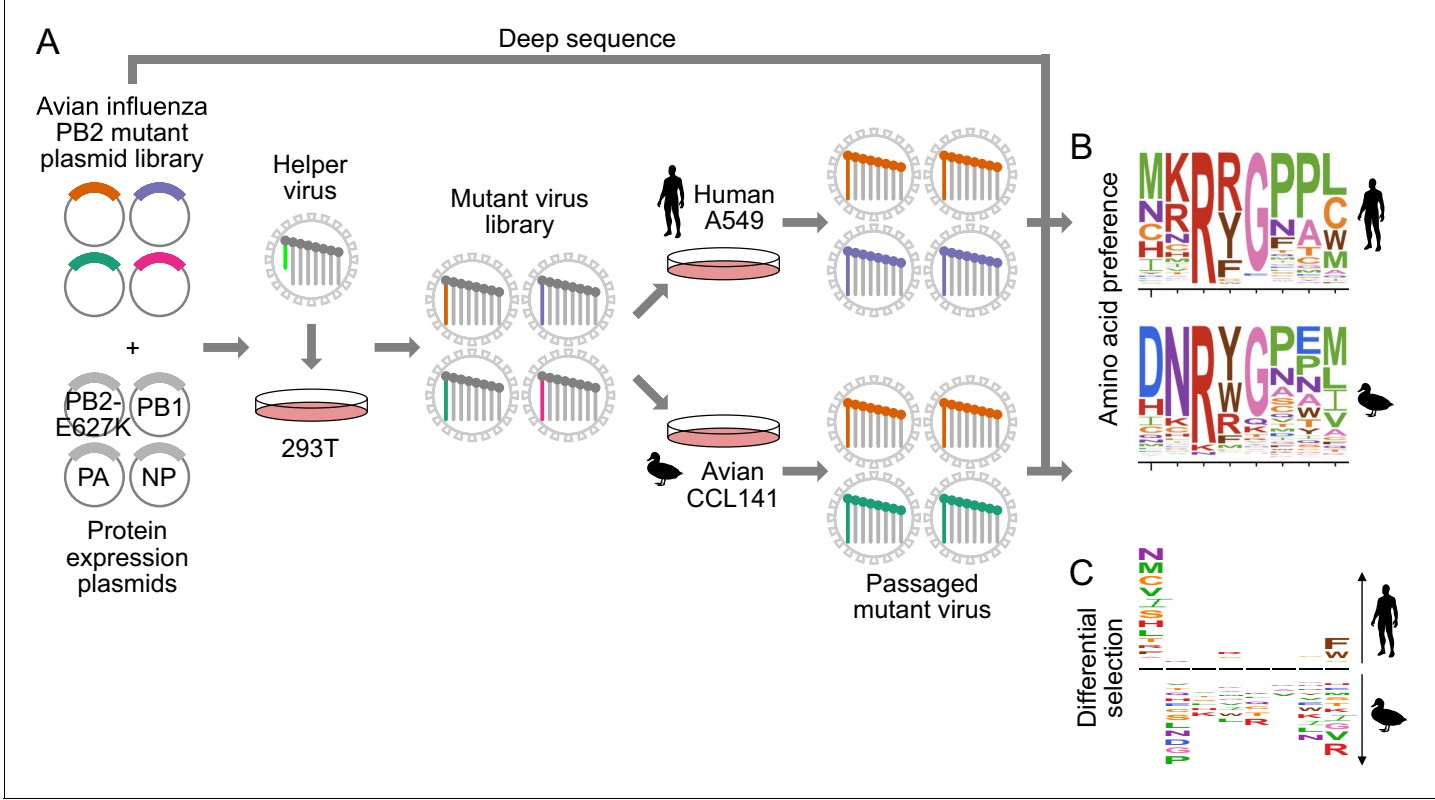

**Figure 1.** Deep mutational scanning of avian influenza PB2 in human and avian cells. (**A**) We mutagenized all codons of PB2 from an avian influenza strain. We generated mutant virus libraries using a helper-virus approach, and passaged libraries at low MOI in human (A549) or duck (CCL-141) cells to select for functional PB2 variants. (**B**) We deep sequenced PB2 mutants from the initial mutant plasmid library and the mutant virus library after passage through each cell type. We computed the 'preference' for each amino acid in each cell type by comparing the frequency of each mutation before and after selection. In the logo plots, the height of each letter is proportional to the preference for that amino acid at that site. (**C**) To identify mutations that are adaptive in one cell type versus the other, we computed the differential selection by comparing the frequency of each amino-acid mutation in human versus avian cells. Letter heights are proportional to the log enrichment of the mutation in human versus avian cells. *Figure 1—figure supplement 1* shows the phylogenetic relation of the chosen avian influenza strain to other influenza strains. *Figure 1—figure supplement 2* shows further details of deep mutation scanning experiment. *Figure 1—figure supplement 3* shows relative amplification of full-length PB2 versus PB2-GFP and PB2-deletion gene segments.

DOI: https://doi.org/10.7554/eLife.45079.003

The following figure supplements are available for figure 1:

**Figure supplement 1.** Phylogenetic relationship of PB2 sequence of chosen avian influenza strain to other influenza strains.
DOI: https://doi.org/10.7554/eLife.45079.004

**Figure supplement 2.** Details of deep mutational scanning experiment.
DOI: https://doi.org/10.7554/eLife.45079.005

**Figure supplement 3.** Relative amplification of full-length PB2 versus PB2-GFP and PB2-deletion gene segments.
DOI: https://doi.org/10.7554/eLife.45079.006

## Experimental measurements are consistent with natural selection and known functional constraints on PB2

Our experiments reflect known functional constraints on PB2 (*Figure 2A*, *Figure 2—figure supplement 1*). As expected, the start codon shows a strong preference for methionine in both human and avian cells. PB2's cap-binding function is mediated by a hydrophobic cluster of five phenylalanines (F404, F323, F325, F330, F363), H357, E361, and K376 (*Guilligay et al., 2008*). Phenylalanines are strongly preferred in the hydrophobic cluster in both host cell types, with the exception of site 323, which also tolerates aliphatic hydrophobic residues in human cells (*Figure 2A*). E361 is also strongly preferred in both cell types, as is K376 in the duck cells. A number of other amino acids are tolerated at site 376 in human cells, and at site 357 in both cell types. At site 357, aromatic residues tyrosine,

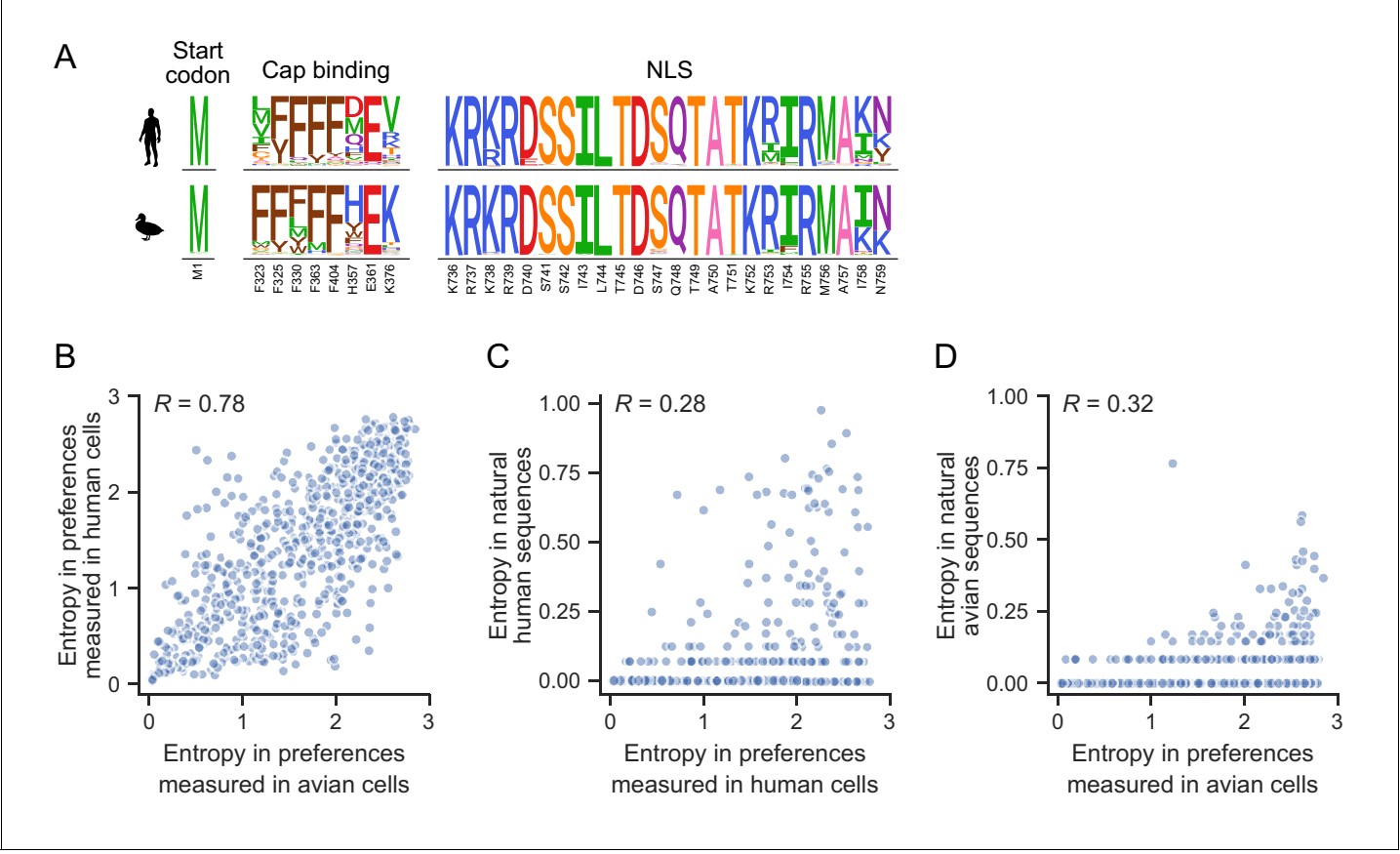

**Figure 2.** Functional constraints on PB2. (**A**) The amino acid preferences measured in human and avian cells for key regions of PB2: the start codon, sites involved in cap-binding, and sites comprising the nuclear localization sequence (NLS). The height of each letter is proportional to the preference for that amino acid at that site. Known critical amino acids are generally strongly preferred in both cell types. (**B**) Correlation of the site entropy of the amino-acid preferences measured in each cell type. (**C**) Sites of high variability (as measured by entropy) in natural human influenza sequences occur at sites of high entropy as experimentally measured in human cells. (**D**) Sites with high variability in natural avian influenza sequences occur at sites of high entropy as experimentally measured in duck cells. *Figure 2—figure supplement 1* shows the complete map of amino acid preferences as measured in human and avian cells. Preferences (as well as mutation effect and differential selection for all mutations as calculated for *Figure 3*) are in *Figure 2— source data 1*.

DOI: https://doi.org/10.7554/eLife.45079.007

The following source data and figure supplement are available for figure 2:

**Source data 1.** Preference, mutation effect, and differential selection results for all mutations.
DOI: https://doi.org/10.7554/eLife.45079.009
**Figure supplement 1.** Complete map of amino acid preferences measured in human and avian cells.
DOI: https://doi.org/10.7554/eLife.45079.008

tryptophan, and phenylalanine are preferred in addition to histidine, consistent with previous observations that the H357W substitution enhances binding to the m7GTP base (*Guilligay et al., 2008*). Finally, the two motifs comprising the C-terminal bipartitite nuclear import signal, 736-KRKR-739 and 752-KRIR-755 (*Tarendeau et al., 2007*), are strongly and similarly preferred in both host cell types. Thus, our experimentally measured preferences largely agree with what is known about PB2 structure and function, and further suggest that functional constraints at these critical sites are similar in both human and avian cells.

To more broadly investigate whether functional constraints are similar between both cell types across the entire PB2 protein, we computed the entropy of the amino acid preferences at each site. A larger site entropy indicates a higher tolerance for mutations at that site. Site entropies are well correlated between cell types (*Figure 2B*, R = 0.78), indicating that sites are usually under similar

functional constraint in both cell types. These protein-wide measures of mutational tolerance are also consistent with natural sequence variation. Natural variation is generally low in natural avian sequences, probably because influenza A virus is highly adapted to avian hosts so there is little pressure for additional adaptation. Natural variation is generally higher in natural human sequences, likely because of increased genetic diversity generated as a result of directional selection to adapt to the human host (*dos Reis et al., 2011*), and diversifying selection to escape immune selection on PB2-derived T-cell epitopes (*Assarsson et al., 2008*). Sites that are highly variable among publicly available natural influenza sequences tend to also be ones that we experimentally measured to be mutationally tolerant (*Figure 2C,D*).

## Identification of human-adaptive mutations

To identify mutations that are adaptive in human versus avian cells, we quantified the host-specific effect of each mutation using two different metrics. The first metric, differential selection, quantifies how much a mutation is selected in one condition versus another (*Doud et al., 2017*). Differential selection is computed by taking the logarithm of the relative enrichment of the mutation relative to the wild-type residue in human versus avian cells (*Figure 1C*, *Figure 3—figure supplement 1A*, *Figure 2—source data 1*). Differential selection greater than zero indicates that a mutation is relatively more favorable in human than avian cells.

To test if differential selection accurately identifies host-specific mutations, we asked if a set of 25 previously experimentally verified human- or mammalian-adaptive mutations (*Figure 3—source data 1*) have differential selection values greater than zero. Indeed, most of these previously characterized mutations had positive differential selection values, as expected for human-adaptive mutations (*Figure 3A*). In contrast, all other mutations have a distribution of differential selection values centered around zero.

However, there are many previously uncharacterized mutations that have differential selection values similar to or greater than those of known human-adaptive mutations (*Figure 3A*). Of course, differential selection only quantifies the extent to which a mutation is more beneficial in human than avian cells. But importantly, for a mutation to be truly adaptive, it must also be more beneficial than the wild-type amino acid in human cells. To quantify each mutation's effect relative to wild type in each cell type, we computed the logarithm of the ratio of preferences of the mutant versus wild-type amino acid (*Figure 3B*, *Figure 2—source data 1*). Mutation effect values greater than zero indicate that a mutation is more preferred than the wild-type residue.

We identified top experimentally adaptive mutations using both differential selection and mutation effect metrics (*Figure 3B*, *Figure 3—figure supplement 1B*). We focused on the 34 mutations most adaptive in our human cell selection (differential selection >1.5 and mutational effect in human cells > 1). Among these 34 mutations, only one (D701N) has already been described as human adaptive. The E627K mutation is favored in human cells in our experiments, though it is not in this set of top 34 mutations. However, two other mutations at this site (E627C and E627S) are among the top 34 mutations (*Figure 3C*). S627 is naturally encoded by bat influenza and, in the context of bat influenza polymerase, supports high polymerase activity in human cells (*Tong et al., 2013*; *Tong et al., 2012*). In fact, it appears that many mutations at site 627 are human adaptive, with the exception of E627D. These observations are consistent with prior findings that a wide range of amino acid residues can be accommodated at site 627, and in fact improve polymerase activity in human HEK293T cells over the consensus avian wild-type glutamic acid (*Chin et al., 2014*). We additionally identified 42 mutations as adaptive in avian cells (differential selection <-2 and mutational effect in avian cells > 1), and seven mutations that are more favorable than the wild-type amino acid in both cell types (mutational effect >1 in both human and avian cells).

From these top adaptive mutations identified in our deep mutational scanning, we chose 26 for experimental validation. Specifically, we chose 18, four, and three mutations adaptive in human, avian, or both cell types, respectively (*Figure 3C,D*). We prioritized mutations that had consistent measurements across biological replicates. When there were multiple strongly adaptive mutations at a site, we chose just one mutation at that site to test mutations across more sites. Finally, we also validated additional mutations of particular interest, such as the only mutation at site 627 (E627D) that appeared to be favored in avian over human cells (negative differential selection), and mutations observed in avian-to-human transmission of H7N9 influenza (see below for more details).

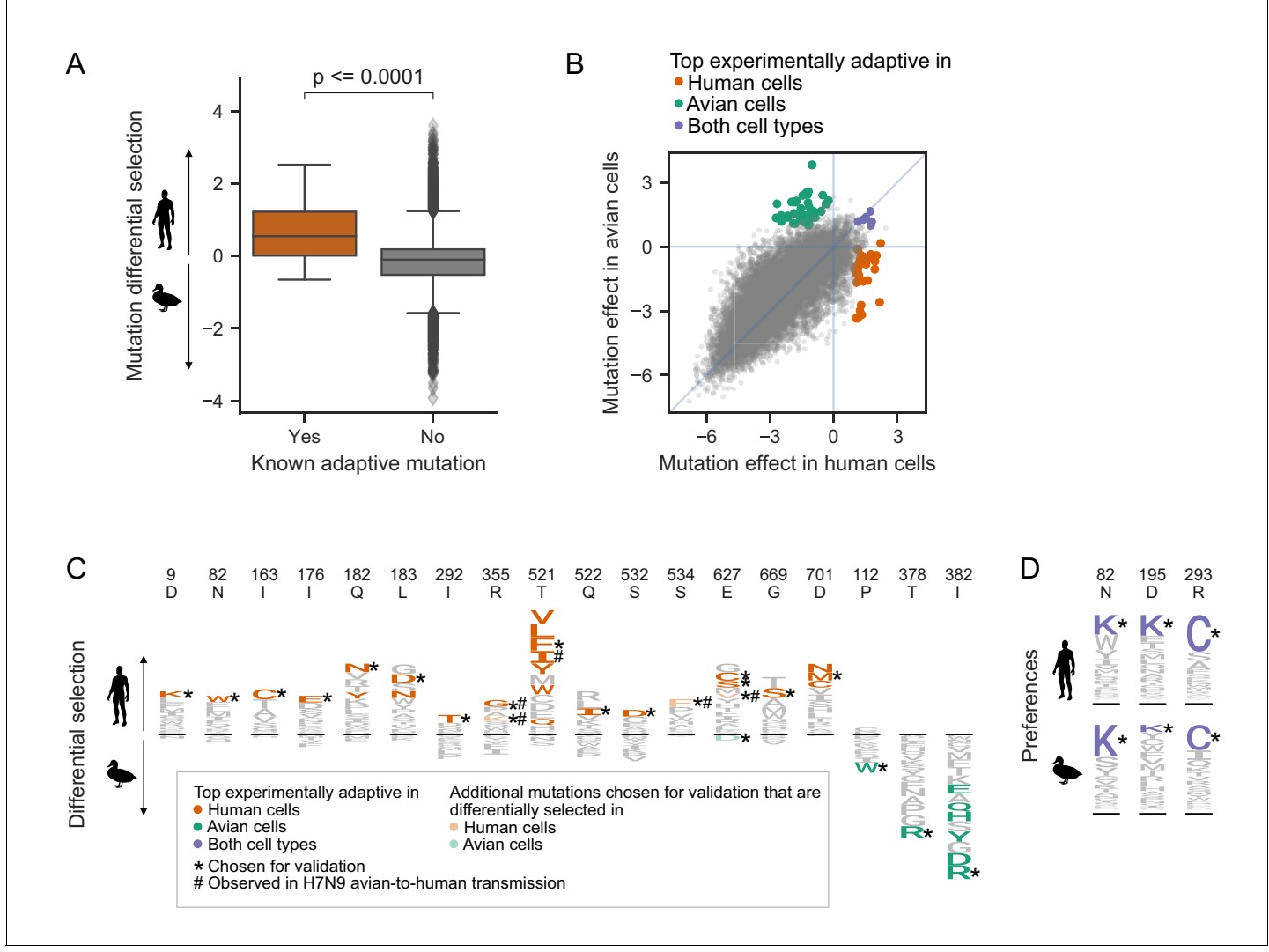

**Figure 3.** Deep mutational scanning identifies known and novel host-adaptive mutations. (A) Distribution of experimentally measured differential selection for previously characterized human adaptive mutations and all other possible mutations to PB2. Positive differential selection means a mutation is favored in human versus avian cells. (B) Scatterplot of each mutation's effect in human versus avian cells, showing the top adaptive mutations identified in the deep mutational scanning. (C) Logoplots showing the differential selection at the sites of mutations that we chose for functional validation. The height of each letter above the line indicates how strongly it was selected in human versus avian cells. Top adaptive mutations are colored in orange (human-adaptive) or green (avian-adaptive). Mutations chosen for functional validation are indicated by an asterisk(*). Additional mutations chosen for validation are colored light orange (differentially selected in human) or light green (differentially selected in bird). Mutations observed in H7N9 avian-to-human transmission are indicated by #. Note that not all mutations with high differential selection in human versus avian cells are classified as top adaptive mutations because we also filtered for mutations that are substantially beneficial relative to wildtype. (D) Logoplots showing amino acid preferences at sites we chose for functional validation. Top mutations beneficial in both human and avian cells are colored purple. Mutations chosen for validation are indicated by *. *Figure 3—figure supplement 1* shows the complete map of differential selection in human versus avian cells. Catalog of previously described human/mammalian adaptive mutations are in *Figure 3—source data 1*.

DOI: https://doi.org/10.7554/eLife.45079.010

The following source data and figure supplement are available for figure 3:

**Source data 1.** Catalog of previously described human/mammalian adaptive mutations.
DOI: https://doi.org/10.7554/eLife.45079.012

**Figure supplement 1.** Complete map of differential selection in human versus avian cells.
DOI: https://doi.org/10.7554/eLife.45079.011

## Human-adaptive mutations identified in deep mutational scanning improve polymerase activity and viral growth in human cells

The main function of the influenza polymerase is to transcribe and replicate the viral genome. We quantified the effect of mutations on polymerase activity using a minigenome assay which measures transcription of an engineered viral RNA encoding GFP by reconstituted influenza polymerase. To test whether the results of our deep mutational scanning are generalizable to human cells beyond the A549 cell line used in the scanning, we performed the minigenome polymerase activity assay in HEK293T as well as A549 cells.

Almost all the putative human-adaptive mutations identified in the deep mutational scanning improved transcriptional activity in human cells relative to the wild type or a synonymous mutant (*Figure 4A,B*, *Figure 4—source data 1*). Mutations that did not improve transcriptional activity retained at least wild-type activity. The effect of mutations in both human cell lines were remarkably consistent, suggesting that our deep mutational scanning yielded results that generalize across human cells.

In contrast, all but one of the putative avian-adaptive mutations decreased transcriptional activity in human cells compared to wild type, as expected (*Figure 4A,B*). The one mutation that did not decrease transcriptional activity, I382D, had comparable activity to wild type. Finally, mutations that are putatively adaptive in both human and avian cells had modestly improved or comparable transcriptional activity in human cells compared to wild type.

We also tested the effect of some of the mutations on viral growth to capture any effects of mutations beyond transcriptional activity. We selected some mutations that increased transcriptional activity, and others that had no effect of transcriptional activity. Our rationale in these choices was to determine if mutations that were identified in our screen but did not increase transcriptional activity still resulted in improved viral growth in human over avian cells. For each competition, we infected human (A549) and avian (CCL-141) cells with mutant and wild-type viruses mixed at a 1:1 ratio of transcriptionally active particles as determined by flow cytometry for HA expression in infected cells. We then measured the frequencies of mutant to wild-type virus by deep sequencing. To measure kinetics of viral genome replication from a single round of infection, we infected cells at MOI of 0.1, collected samples at 10 hr post infection, and sequenced vRNA from cellular extract. To measure multi-cycle replication kinetics, we infected cells at MOI of 0.01, collected samples at 48 hr post infection, and sequenced vRNA from the supernatant. At the end of the infection, we calculated the ratio of mutant to wild-type virus in human cells divided by the same ratio in avian cells. Because this quantity is a ratio of ratios, it corrects for any possible deviations from a 1:1 ratio of infectious particles in the initial inoculum. Therefore, ratios greater than one indicates that the mutation confers a relative benefit for viral growth in avian versus human cells.

Almost all putative human-adaptive mutations identified in the deep mutational scanning improved growth in human over avian cells, as reflected by an increase in the ratio of mutant to wild type in human cells versus avian cells over the time course of the competition (*Figure 4C*, *Figure 4—source data 2*). Of the putative human-adaptive mutations that did not improve polymerase activity, one mutation, R355G, improved growth in human over avian cells at both 10 and 48 hr post-infection. An additional mutation, R355K, slightly improved growth in human cells by 48 hr post-infection. These two mutations may therefore confer a human-specific growth benefit due to some mechanism other than polymerase activity. As expected, both putative avian-adaptive mutations resulted in poorer growth in human versus avian cells. Finally, the N82K mutation that is putatively adaptive in both human and avian cells resulted in comparable growth in both human and avian cells, as expected.

Thus, our deep mutational scanning identified numerous previously undescribed PB2 mutations that improve polymerase activity or viral growth in human cells. In addition, it also identified an intriguing small set of mutations that enhance viral growth but not polymerase activity in human cells.

## Human-adaptive mutations cluster in regions of PB2 that are potentially important for host adaptation

The additional human-adaptive mutations we identified may improve PB2's ability to interact with important human cell factors. To identify potential interfaces for such interactions, we mapped the

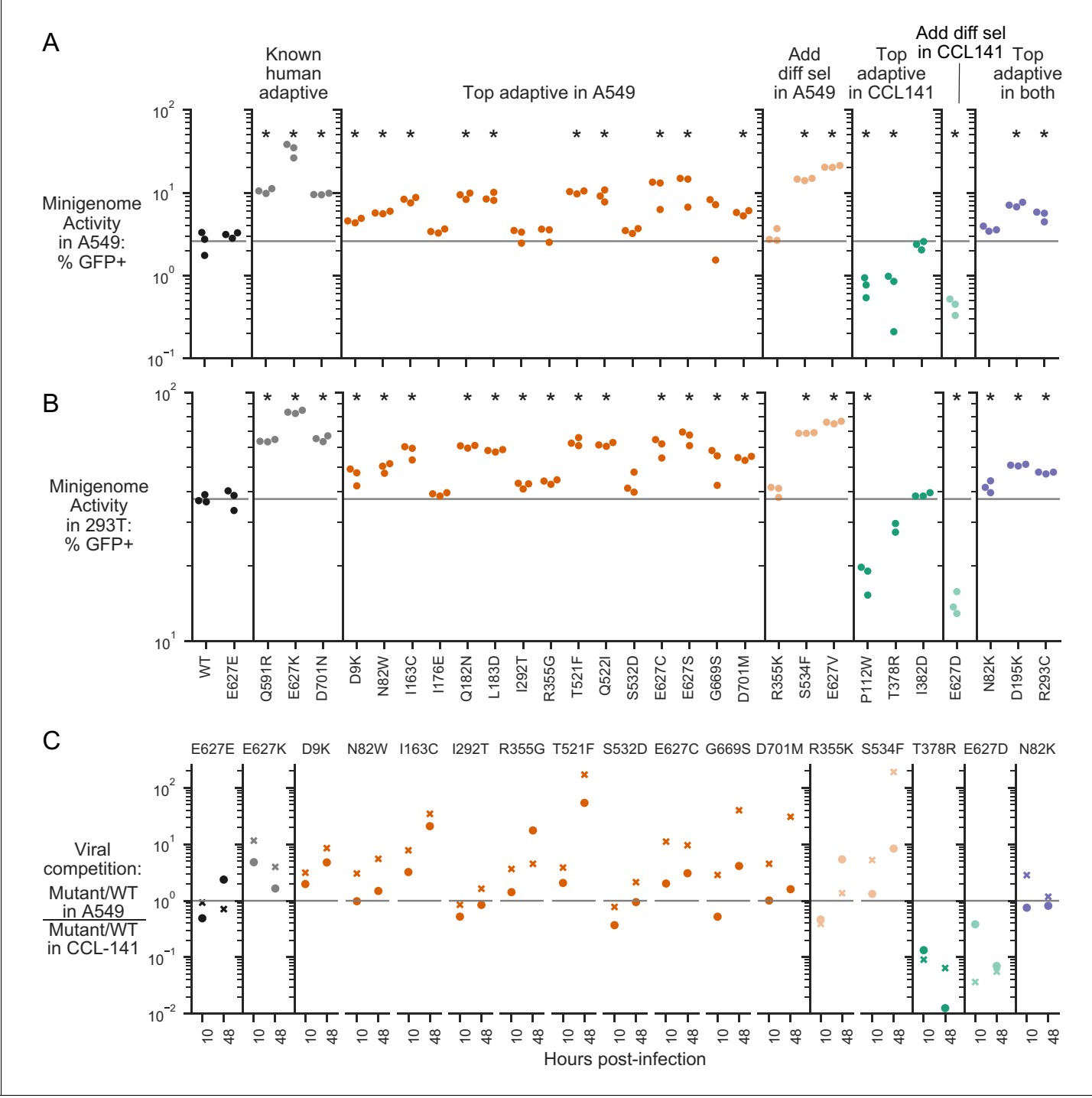

**Figure 4.** Validation of top experimentally adaptive mutations. The polymerase activity of selected PB2 mutants as measured using minigenome assays in A549 (**A**) and HEK293T (**B**) human cells. The mutations chosen for characterization include previously known human adaptive mutations, top adaptive mutations identified by our deep mutational scanning (orange = human adaptive, green = avian adaptive), and additional mutations differentially selected in human (light orange) or avian (light green) cells. E627E is a synonymous mutation at site 627 used as a negative control. Minigenome activity is represented as percent of transfected cells that expressed a viral GFP reporter. The gray horizontal line indicates the mean value measured for the wild type avian PB2. Minigenome assays were performed in biological triplicate. Mutations that have significantly different minigenome activity from wild type are indicated by asterisks (unpaired t-test, $p < 0.05$). (**C**) Competition of virus bearing the indicated mutant PB2 against virus with wild-type PB2. For each competition, human A549 and avian CCL-141 cells were infected with mutant and wild-type viruses mixed at a 1:1 ratio of transcriptionally active particles, and the frequency of each variant after viral replication was measured by deep sequencing viral RNA. For samples collected at 10 hr post infection, we infected cells at MOI of 0.1, and sequenced vRNA from cellular extract. For samples collected at 48 hr post

*Figure 4 continued on next page*

*Figure 4 continued*

infection, we infected cells at MOI of 0.01, and sequenced vRNA from the supernatant. The plots show the ratio of the mutant over wild-type variant in A549, divided by the same ratio in CCL-141 cells. A ratio >1 indicates that a viral mutant grows better in human than avian cells. Competition assays were performed in biological duplicate; circle and cross represent replicate experiments. Flow data for minigenome activity and and mutation counts for viral competition are provided in *Figure 4—source datas 1* and *2*.

DOI: https://doi.org/10.7554/eLife.45079.013

The following source data is available for figure 4:

**Source data 1.** Flow cytometry data for minigenome assays.
DOI: https://doi.org/10.7554/eLife.45079.014

**Source data 2.** Mutant frequency data for competition assay.
DOI: https://doi.org/10.7554/eLife.45079.015

sites of top human-adaptive mutations identified in the deep mutational scanning onto the structure of PB2. Many of the sites cluster in regions of PB2 that may play a role in host adaptation (*Figure 5*, *Figure 5—figure supplement 1*).

Most adaptive mutations are surface exposed in at least one of the two conformations of the polymerase we examined (*Figure 5—figure supplement 1A*, relative solvent accessibility >0.2). In the transcription pre-initiation form of the polymerase (PDB: 4WSB) (*Reich et al., 2014*) (*Figure 5A, B*), the experimentally identified sites 532 and 292 are located on the surface near sites of known human-adaptive sites 286, 534, and 535 (*Cauldwell et al., 2013*; *Mänz et al., 2016*) (*Figure 5B*: i). Experimentally identified sites 69 and 82 are located on the surface close to sites 64 and 81 (*Figure 5B*: ii), which are located near the template exit channel and were recently shown to modulate generation of mini viral RNAs that act as innate-immune agonists (*Te Velthuis et al., 2018*). Experimentally identified site 698 is located near known sites 701 and 702 (*Gabriel et al., 2005*) (*Figure 5B*: iii, *Figure 5D*: i). Experimentally identified mutations are also located in regions not yet identified to be important for host-adaptation: we find a cluster of experimentally identified sites on the surface of the PB2-N terminal domain (169, 176, 183, 190, *Figure 5B*: iv), partially occluded by the flexible PA endonuclease domain in the transcription pre-initiation structure. Experimentally identified sites 163 and 182 are not surface exposed, but are buried right underneath the cluster of four sites.

Almost all sites that are not highly surface exposed in the transcription pre-initiation conformation of the polymerase become exposed upon conformational rearrangement of the polymerase from the transcription pre-initiation form to the apo form (PDB: 5D98) (*Hengrung et al., 2015*). The experimentally identified sites 521 and 522 which face the product exit channel, site 355 which faces the core of the polymerase, and site 669 which faces PB1 in the transcription pre-initiation structure (*Figure 5B*: v – vii), are more fully surface-exposed in the apo structure (*Figure 5C,D*: ii – iv). However, some sites remain inaccessible: site 156 faces the internal core of the polymerase in both transcription pre-initiation and apo forms (*Figure 5B*: viii, *Figure 5D*: v). Therefore, with only a few exceptions, the human-adaptive mutations cluster in patches on the surface of PB2. Taking a comprehensive approach has therefore allowed us to map surfaces of PB2 that might mediate host-interactions.

Next, we asked if host-adaptation mutations occur at known interaction interfaces with host proteins. One known interacting host protein is importin-α, which mediates nuclear import of PB2 and has been proposed to have a role in viral transcription and replication (*Resa-Infante et al., 2008*; *Tarendeau et al., 2007*). PB2 of avian viruses uses importin-α3 in human cells, whereas PB2 of mammalian-adapted viruses uses importin-α7 (*Gabriel et al., 2011*; *Gabriel et al., 2008*). We mapped total positive differential selection on each site of PB2 (see legend for *Figure 5E*), and asked how this selection on PB2 relates to its interaction with importin-α. Sites on PB2 that interact with the major and minor NLS-binding surfaces of importin-α (*Pumroy et al., 2015*; *Tarendeau et al., 2007*) generally have low differential selection, indicating that host-adaptation mutations do not occur at these sites (*Figure 5E*; PDB: 4UAD) (*Pumroy et al., 2015*). This is expected, since all importin-α isoforms share an invariant NLS-binding surface. However, adjacent PB2 sites have higher differential selection (*Figure 5E*: i-iii). Some of these PB2 sites are in close proximity to regions of importin-α

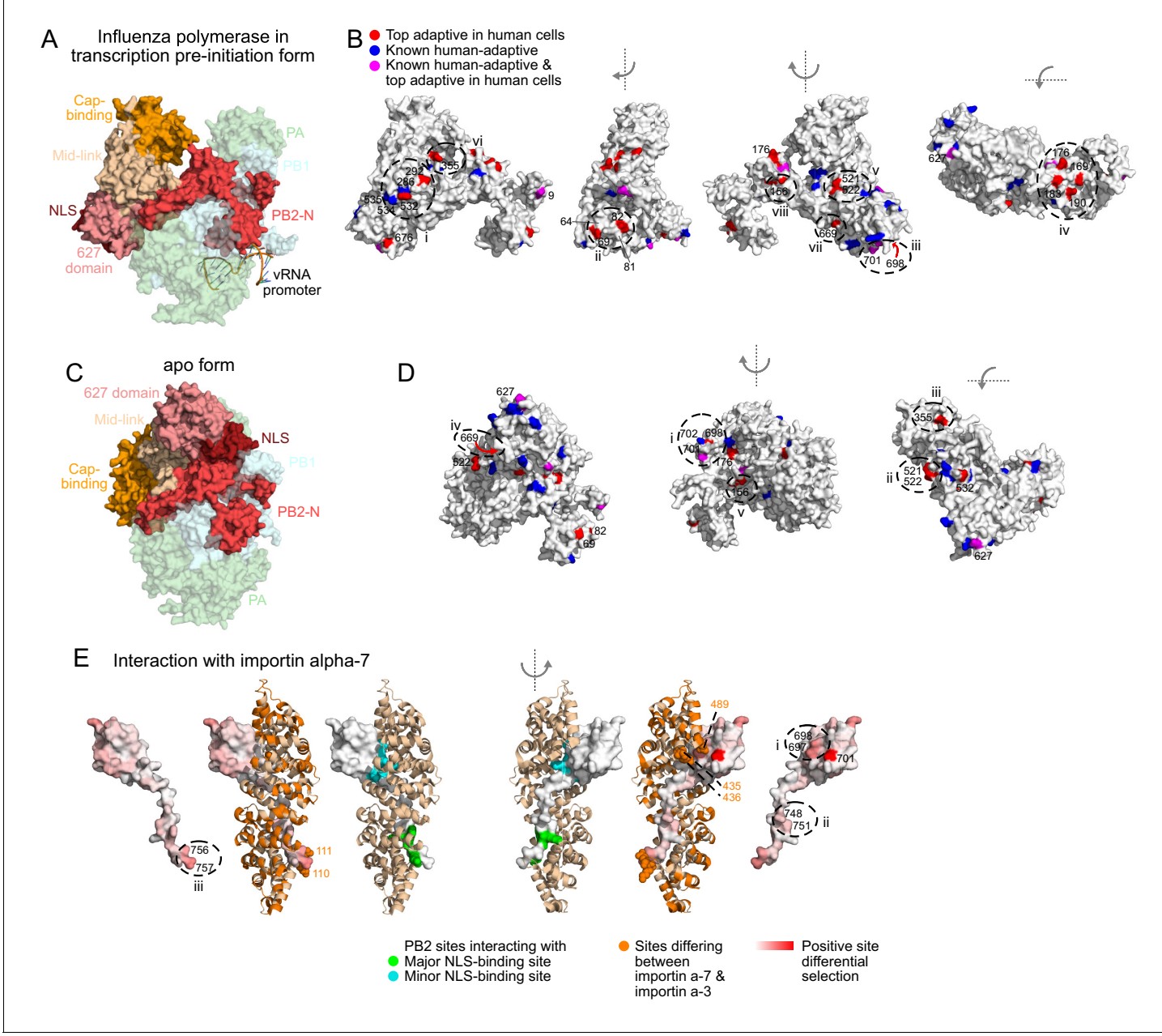

**Figure 5.** Locations of top human-adaptive mutations on the structure of the influenza polymerase. Overall structure of the influenza polymerase complex comprising PB2, PB1 and PA in (**A, B**) the transcription pre-initiation form (PDB: 4WSB) and (**C, D**) the apo form (PDB: 5D98). PB2 domains defined as in *Pflug et al. (2017)*. (**B, D**) Sites of top human-adaptive mutations identified by deep mutational scanning are shown in red on the PB2 subunit of the structure. Sites of previously experimentally verified human-adaptive mutations are in blue (25 sites as listed in *Figure 3—source data 1*). Sites identified by deep mutational scanning and which were also previously known are in purple. A subset of sites are labeled and/or circled for referencing in the main text, to indicate surfaces that might mediate host-interactions. Similar results are obtained if we instead analyze the structures in terms of a continuous variable representing the extent of human-specific adaptation at each site (*Figure 5—figure supplement 1B, C*). (**E**) Structure of PB2 C-terminal fragment co-crystalized with importin-α7 (PDB: 4UAD). Sites on PB2 interacting with major and minor NLS binding surfaces of importin-α7 are in green and cyan respectively. Importin-α7 is depicted in ribbon form in tan. We used the deep mutational scanning to define a continuous variable indicating the extent of host-specific adaptation at each site of PB2. Specifically, for each site, we computed the positive site differential selection by summing all positive mutation differential selection values at that site (i.e., the total height of the letter stack in the positive direction in logoplots such as in *Figure 3D*). We mapped this differential selection onto the PB2 C-terminal fragment in red; PB2 sites with high differential selection are numbered. Regions of importin-α7 that differ from importin-α3 are colored in orange, those near PB2 sites with high differential selection are shown as spheres. For all structures, the avian influenza (S009) PB2 amino acid sequence was mapped onto the PB2 chain by one-2-one threading using Phyre2 (*Kelley et al., 2015*) (Confidence in models for 4WSB, 5D98, and 4UAD are 100%, 100%, and 99% respectively). Sites are numbered

*Figure 5 continued on next page*

*Figure 5 continued*

according to the S009 PB2 sequence. *Figure 5—figure supplement 1* shows relative solvent accessibility of human-adaptive mutations, as well as positive site differential selection mapped onto structures of influenza polymerase.

DOI: https://doi.org/10.7554/eLife.45079.016

The following figure supplement is available for figure 5:

**Figure supplement 1.** Solvent accessibility of sites of human-adaptive mutations, and positive site differential selection mapped onto structures of influenza polymerase.

DOI: https://doi.org/10.7554/eLife.45079.017

that differ between the $\alpha-7$ and $\alpha-3$ isoforms (*Figure 5D*: i, iii), suggesting that adaptation at these PB2 sites affects importin-$\alpha$ usage.

PB2 also interacts with the C-terminal domain of RNA polymerase II, and this interaction is proposed to stabilize the polymerase in the transcription-competent conformation (PDB: 6F5O) (*Serna Martin et al., 2018*). Similar to what we observe with importin-$\alpha$, PB2 sites thought to directly interact with RNA polymerase II tend to have low differential selection, whereas adjacent PB2 sites have higher differential selection (*Figure 5—figure supplement 1D*). Thus, it appears that host adaptation may involve mutations at sites adjacent to the core residues that directly interact with host proteins.

## Experimentally defined human-adaptive mutations are enriched in avian-to-human transmission of H7N9 influenza

A challenge in the surveillance of non-human influenza and assessment of pandemic risk is determining which of the many mutations that occur during viral evolution are human-adaptive (*Lipsitch et al., 2016*; *Russell et al., 2014*). We investigated whether our experimental measurements can identify host-adaptation mutations that occur during the actual transmission of avian influenza to humans.

Avian H7N9 influenza viruses have recently caused a large number of sporadic human infections (*Su et al., 2017*). We examined mutations occurring during the evolution of H7N9 viruses that have jumped from avian to human hosts to determine whether they were enriched for changes predicted by our deep mutational scanning to be human adaptive. First, we constructed a phylogeny of H7N9 PB2 sequences, inferred ancestral sequences for all internal nodes, and assigned mutations to specific branches of the phylogenetic tree (*Figure 6A*, *Figure 6—figure supplement 1–5*). We then classified each mutation on the phylogenetic tree as 'avian' or 'human' based on whether it occurred on a branch connecting two avian isolates, or on a branch leading to a human isolate respectively (*Figure 6—source data 1*). As human infections by H7N9 are evolutionary dead-ends, mutations occurring during human infections should appear immediately proximal to human isolates in the phylogeny, while mutations occurring during bird infections will be ancestral.

We next asked whether mutations occurring in human hosts had higher differential selection values in our deep mutational scanning than mutations in avian hosts. Indeed, human mutations more often had high differential selection (a value >0.5) than avian mutations (Fisher's exact test, p=2.69e-7) (*Figure 6B*). The H7N9 human mutations with differential selection >0.5 include the well-studied human-adaptive mutations 627K and 701N. Indeed, these two mutations make up the majority of H7N9 human mutations with high differential selection. But we also identified a number of other mutations with high differential selection that occurred at least four independent times in jumps of H7N9 influenza into humans: 627V, 534F, 355K, and 521I (*Figure 6A*, *Figure 6—figure supplement 1*–5, 6B, 3D), only one of which has been previously characterized (E627V, *Taft et al., 2015*). A second mutation at site 355 (355G) with high differential selection also occurs during jumps of H7N9 influenza into humans. Thus, our deep mutational scanning identifies both previously characterized and novel mutations that occur in natural avian-to-human transmission of influenza.

## Most human-adaptive mutations are not accessible by single nucleotide substitutions

Our deep mutational scanning identifies many human-adaptive mutations. Why do we not observe all of them in nature? One possible explanation is that some of these mutations are inaccessible by

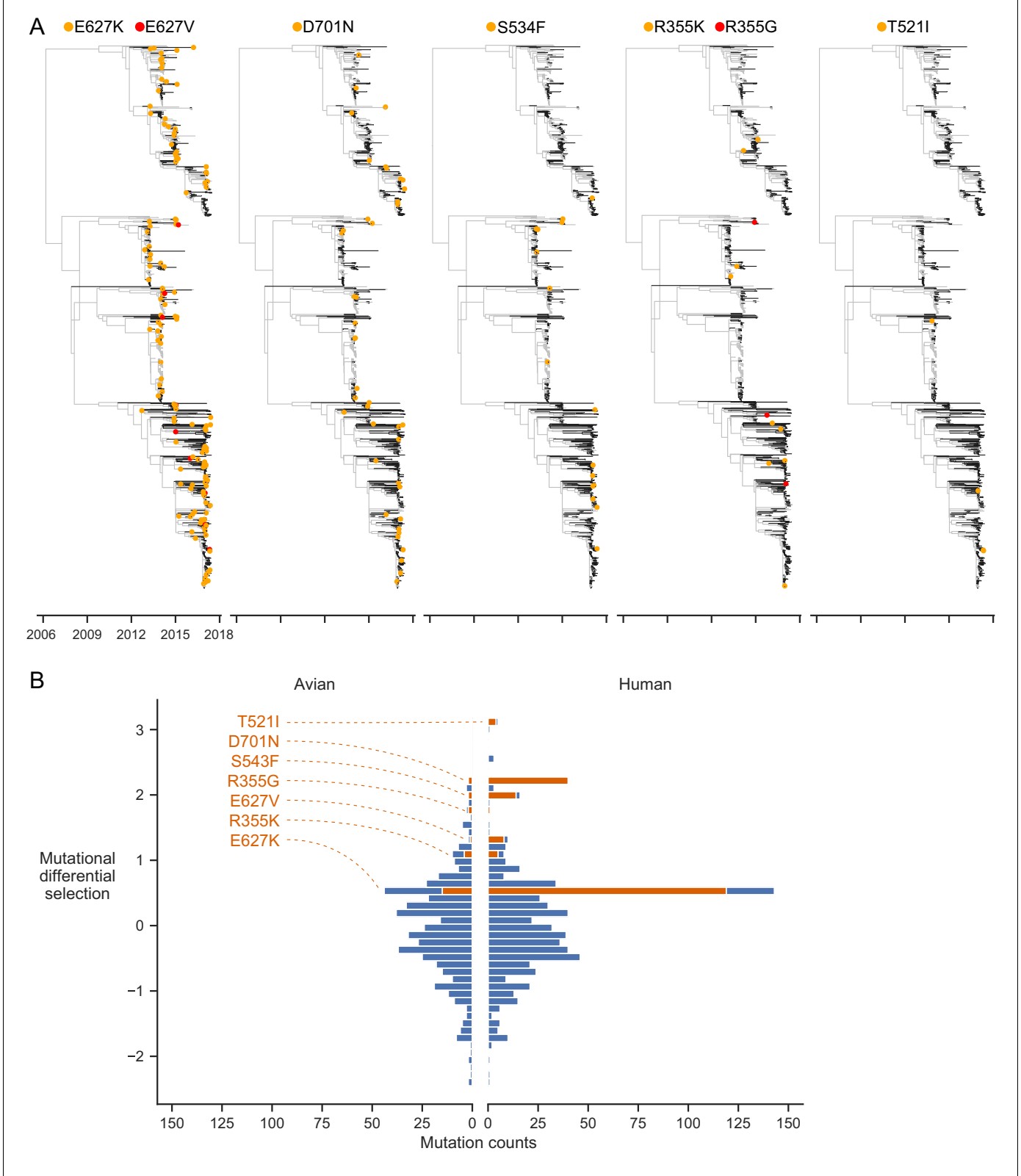

**Figure 6.** Experimentally identified human-adaptive mutations are enriched in avian-human transmission of H7N9 influenza. (A) Phylogeny of H7N9 influenza PB2 sequences. Branches in human and avian hosts are colored black and gray respectively. Orange or red dots indicate where a mutation was inferred to have occurred. Branch lengths are scaled by annotated and inferred dates of origin of each sequence. (B) Distribution of experimentally measured differential selection values for all mutations occurring during H7N9 evolution in human and avian hosts. A positive differential selection value

*Figure 6 continued on next page*

*Figure 6 continued*

means that our experiments measured the mutation to be beneficial in human versus avian cells. A subset of top differentially selected mutations that occur frequently are labeled and plotted in orange. Enlarged phylogenetic trees are in *Figure 6—figure supplement 1–5*. Counts of mutations identified in phylogenetic analysis are in *Figure 6—source data 1*. Mutations plotted in each bin of the histogram are in *Figure 6—source data 2*.
DOI: https://doi.org/10.7554/eLife.45079.018

The following source data and figure supplements are available for figure 6:

**Source data 1.** H7N9 human and avian mutation counts.
DOI: https://doi.org/10.7554/eLife.45079.024
**Source data 2.** H7N9 human and avian mutation differential selection values and counts in each histogram bin.
DOI: https://doi.org/10.7554/eLife.45079.025
**Figure supplement 1.** Phylogeny of H7N9 influenza PB2 sequences showing where mutations at site 627 were inferred to have occurred.
DOI: https://doi.org/10.7554/eLife.45079.019
**Figure supplement 2.** Phylogeny of H7N9 influenza PB2 sequences showing where mutations at site 701 were inferred to have occurred.
DOI: https://doi.org/10.7554/eLife.45079.020
**Figure supplement 3.** Phylogeny of H7N9 influenza PB2 sequences showing where mutations at site 534 were inferred to have occurred.
DOI: https://doi.org/10.7554/eLife.45079.021
**Figure supplement 4.** Phylogeny of H7N9 influenza PB2 sequences showing where mutations at site 355 were inferred to have occurred.
DOI: https://doi.org/10.7554/eLife.45079.022
**Figure supplement 5.** Phylogeny of H7N9 influenza PB2 sequences showing where mutations at site 521 were inferred to have occurred.
DOI: https://doi.org/10.7554/eLife.45079.023

single nucleotide substitution from existing sequences, and are therefore less likely to arise during natural evolution (*Fragata et al., 2018*).

We examined if accessibility by single nucleotide substitution imposes constraint on which human-adaptive mutations arise in nature. To do so, we calculated the mean nucleotide substitutions required to access known human-adaptive mutations from all avian influenza PB2 sequences collected in the past three years. This mean number of nucleotide substitutions can range from less than one (if the mutation is already present in some avian PB2 sequences) to three (if the mutation requires three nucleotide changes from all avian PB2 sequences). The majority of previously characterized human-adaptive mutations are accessible by single nucleotide substitutions from avian PB2 sequences (*Figure 7*), suggesting that these mutations have already been characterized because they readily occur in the context of current avian influenza viruses. In contrast, most of the top human-adaptive mutations identified in our deep mutational scanning require multiple nucleotide substitutions from current avian PB2 sequences (*Figure 7*). Therefore, many of the novel human-adaptive mutations uncovered by our experiment have probably not been previously identified because they are evolutionarily inaccessible from current avian influenza sequences.

The importance of evolutionary accessibility is especially obvious if we examine the adaptive mutations that actually occur in natural influenza virus evolution. Of the five top human adaptive mutations accessible by single nucleotide substitution (*Figure 7*), three have occurred repeatedly during recent transmissions of H7N9 influenza to humans (355G, 521I, 701N; see *Figure 6A*). Thus, it may be possible to combine deep mutational scanning measurements of phenotypes with analyses of evolutionary accessibility to anticipate which mutations are most likely to arise in the context of any given starting sequence (*Fragata et al., 2018*).

## Discussion

We have measured how all single amino-acid mutations to an avian influenza PB2 affect viral growth in both human and avian cells. Our results separate the constraints on PB2 into those that are maintained across cells from diverse species versus those that are specific to human or avian cells. The vast majority of sites are under extremely similar constraint in human and avian cells, including at residues already known to be critical for PB2 function. Layered upon this conserved constraint are mutations with host-specific effects. Our approach therefore represents a powerful strategy for mapping viral determinants of cross-species transmission.

The viral determinants of influenza host specificity in the PB2 protein have been intensely studied for decades. Earlier studies addressing this question focused on mutations that have fixed during

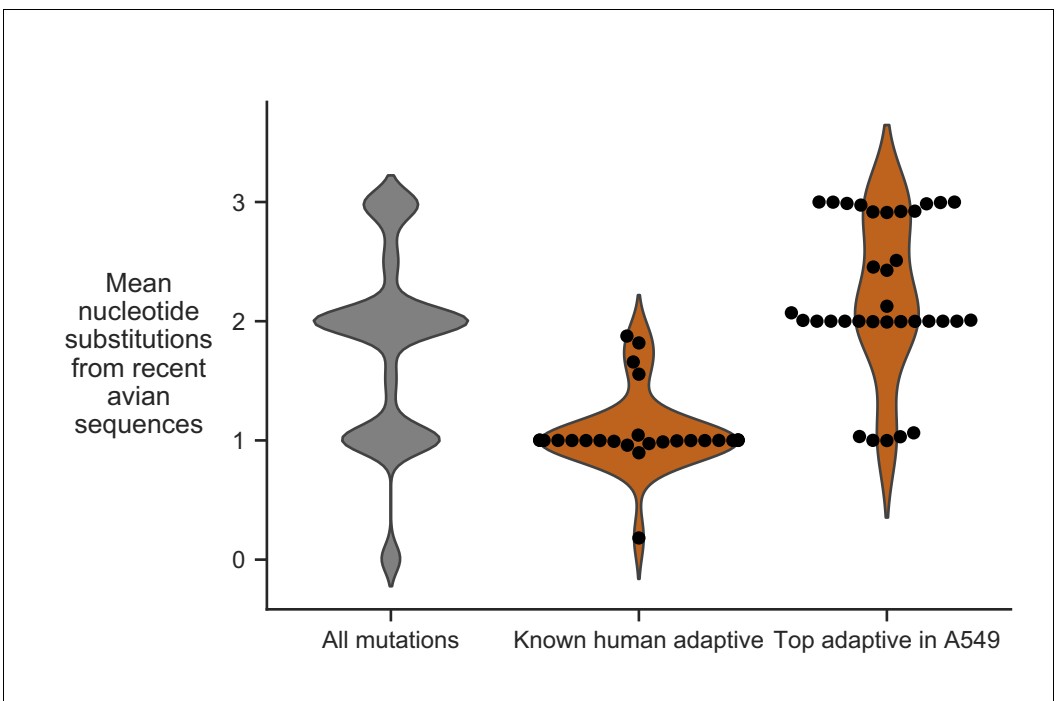

**Figure 7.** Evolutionary accessibility of mutations from current avian influenza PB2 sequences. Distribution of mean nucleotide substitutions required to access all amino-acid mutations, previously characterized human-adaptive mutations, and top human-adaptive mutations identified in our deep mutational scanning. Mean nucleotide substitution is calculated by averaging over all avian influenza PB2 sequences collected from 2015 to 2018. Most previously characterized human-adaptive mutations are accessible by single nucleotide substitution, whereas many of the new adaptive mutations that we identified require multiple nucleotide substitutions. Mean nucleotide substitutions for each mutation are in *Figure 7—source data 1*.

DOI: https://doi.org/10.7554/eLife.45079.026

The following source data is available for figure 7:

**Source data 1.** Mean nucleotide substitutions from avian sequences of all mutations.

DOI: https://doi.org/10.7554/eLife.45079.027

viral adaptation in nature or in the lab (*Bussey et al., 2010*; *Cauldwell et al., 2013*; *Chen et al., 2006*; *Finkelstein et al., 2007*; *Gabriel et al., 2005*; *Hu et al., 2017*; *Hu et al., 2014*; *Kim et al., 2010*; *Mänz et al., 2016*; *Mehle and Doudna, 2009*; *Miotto et al., 2008*; *Mok et al., 2011*; *Naffakh et al., 2000*; *Tamuri et al., 2009*; *Yamada et al., 2010*; *Zhou et al., 2011*). Recent studies have sampled more mutations by random mutagenesis at key sites such as positions 627, 701, and 702 of PB2 (*Chin et al., 2014*; *Chin et al., 2017*), or error prone PCR across the genome (*Taft et al., 2015*). We can now place these previously characterized mutations in the context of a complete map, thus revealing how selection at molecular interfaces and evolutionary accessibility shapes influenza's adaptation to humans.

Examination of our maps of differential selection at known molecular interfaces, such as with importin-α (*Pumroy et al., 2015*; *Tarendeau et al., 2007*) and RNA polymerase II (*Serna Martin et al., 2018*) suggest an interesting hypothesis: host-adaptive mutations are located adjacent to but not at core residues that directly interact at host proteins, suggesting that host adaptation may involve mutations at sites at the periphery of core interactions. Of course, it remains unclear precisely how any of these new human-adaptive mutations act, though initial validations show that most improve polymerase activity in human cells. However, an intriguing set of mutations such as at site 355 improve viral growth but not polymerase activity in human cells. We speculate that the effects of these mutations on viral growth are mediated by other effects of PB2, such as in modulating the innate-immune response (*Graef et al., 2010*; *Te Velthuis et al., 2018*).

Even though we have focused our analyses mostly on host-adaptive mutations, our measurements are also useful to studies of PB2 function independent of host adaptation. Our complete maps of

amino acid preferences in each cell type provide context for the handful of residues well-known to be critical for PB2 functions. For example, although our data recapitulate most of the known functional constraints on sites previously defined to be critical for mRNA cap-binding (*Guilligay et al., 2008*), they also reveal that some of these sites can tolerate alternate residues. Thus, our complete survey of the mutational constraints at each site complements other functional and structural knowledge.

We also show that our comprehensive experimental measurements can identify human-adaptive mutations that occur during avian-to-human transmission of H7N9 influenza. These measurements therefore help address a fundamental challenge in assessing the risk of potential pandemic influenza virus strains: determining which of the many mutations observed during viral surveillance affect whether a virus will be successful in human hosts (*Lipsitch et al., 2016*; *Russell et al., 2014*). Our high-throughput approach therefore enables phenotypic measurements to keep pace with the challenge of interpreting the many viral mutations that are observed during genotypic surveillance of virus evolution (*Grubaugh et al., 2019*).

However, although some amino-acid mutations we experimentally identified as human-adaptive occur frequently in nature, others have never been observed. In addition, the strengths of differential selection measured in our experiments do not necessarily correspond to observed frequencies of mutations in nature. These apparent inconsistencies highlight the important role of evolutionary accessibility in shaping influenza's host adaptation. First, most human-adaptive mutations observed in natural influenza evolution are accessible by single nucleotide substitution from current avian genotypes, demonstrating how the architecture of the genetic code impacts viral adaptation (*Fragata et al., 2018*). Second, even among single-nucleotide substitutions, transition mutations are about ten-fold more frequent than transversions during influenza replication (*Bloom, 2014*). This might explain why E627K (which requires a G→A transition) is much more frequently observed than E627V (which requires a A→T transversion), despite the latter being more strongly selected in human cells in our experiments. Of course, many additional factors not captured by our experiments can impact which adaptive mutations fix in nature. Integrating our complete maps of the effects of amino-acid mutations with other data that sheds light on evolutionary opportunity, such as nucleotide accessibility, mutation rates, transmission bottlenecks, and environmental and epidemiological factors (*Geoghegan et al., 2016*; *Geoghegan and Holmes, 2017*; *Moncla et al., 2016*; *Olival et al., 2017*; *Peck and Lauring, 2018*; *Varble et al., 2014*) will help us understand how viruses cross species barriers in nature.

# Materials and methods

### Key resources table

| Reagent type (species) or resource | Designation | Source or reference | Identifiers | Additional information |
|---|---|---|---|---|
| Cell line (Homo sapiens) | A549 | ATCC | CCL-185; RRID:CVCL_0023 | |
| Cell line (Homo sapiens) | HEK293T | ATCC | CRL-3216; RRID:CVCL_0063 | |
| Cell line (Canis familiaris) | MDCK-SIAT1 | Sigma-Aldrich | 5071502; RRID:CVCL_Z936 | |
| Cell line (Anas platyrhynchus domesticus) | CCL-141 | ATCC | CCL-141; RRID:CVCL_T281 | |
| Cell line (Canis familiaris) | MDCK-SIAT1-tet-S009-PB2-E627K | this paper | | MDCK-SIAT1 cells expressing S009 PB2-E627K under control of a doxycycline-inducible promoter |
| Recombinant DNA reagent | pHW_noCMV_S009_PB2; pHW_noCMVnoTerm_BsmBI | this paper | | Plasmids for generating mutant plasmid library; see *Supplementary file 1* |

*Continued on next page*

*Continued*

| Reagent type (species) or resource | Designation | Source or reference | Identifiers | Additional information |
|---|---|---|---|---|
| Recombinant DNA reagent | pHW_S009_PB2; pHW_S009_PB1; pHW_S009_PA; pHW_S009_NP | this paper | | Plasmids for generating helper virus; see *Supplementary file 1* |
| Recombinant DNA reagent | HDM_S009_PB2; HDM_S009_PB1; HDM_S009_PA; HDM_S009_NP | this paper | | Plasmids for protein expression of S009 polymerase complex; see *Supplementary file 1* |
| Recombinant DNA reagent | pHH_PB2_S009_flank _99_eGFP_100 | this paper | | Plasmids for generating helper virus; see *Supplementary file 1* |
| Recombinant DNA reagent | pHW184_HA; pHW186_NA; pHW187_M; pHW188_NS | *Hoffmann et al. (2000)* | | |
| Recombinant DNA reagent | pHH-PB1-flank-eGFP | *Bloom et al. (2010)* | | Reporter plasmid for minigenome assay; see *Supplementary file 1* |
| Recombinant DNA reagent | pcDNA3.1_mCherry | this paper | | Transfection control for minigenome assay; see *Supplementary file 1* |
| Recombinant DNA reagent | pSBtet_RP_S009 _PB2_E627K | this paper | | Plasmid for generating PB2-expressing cell line; see *Supplementary file 1* |
| Sequence-based reagent | primers | this paper | | See *Supplementary file 2* |
| Commercial assay or kit | NEBuilder HiFi DNA Assembly Master Mix | New England Biolabs | E2621S | |
| Commercial assay or kit | ElectroMAX DH10B competent cells | Invitrogen | 18290015 | |
| Commercial assay or kit | Rneasy Mini Kit | Qiagen | 74104 | |
| Commercial assay or kit | Accuscript Reverse Transcriptase | Agilent | 200820 | |
| Commercial assay or kit | KOD Hot Start Master Mix | EMD Millipore | 71842 | |
| Commercial assay or kit | QIAamp Viral RNA Mini Kit | Qiagen | 52904 | |
| Commercial assay or kit | SuperScript III | ThermoFisher Scientific | 18080051 | |
| Chemical compound, drug | BioT | Bioland Scientific | B01-01 | |
| Chemical compound, drug | Lipofectamine 3000 | ThermoFisher Scientific | L3000015 | |
| Antibody | H17-L19 | *Gerhard et al. (1981)* | | |
| Software, algorithm | dms_tools2 | https://jbloomlab. github.io/dms_tools2, version 2.3.0 | | |
| Software, algorithm | Jupyter notebooks that perform all steps of analyses | this paper | | See *Supplementary file 3*; https://github.com/ jbloomlab/PB2-DMS |

## Cell lines and media

HEK293T, MDCK-SIAT1, and A549 (ATCC CCL-185) cells were maintained in D10 media (DMEM supplemented with 10% fetal bovine serum, 2 mM L-glutamine, 100 U/ml penicillin, and 100 μg/ml of streptomycin). CCL-141 cells (ATCC CCL-141) were maintained in E10 media (identical to D10 except that EMEM is used in place of DMEM). Cells were grown in WSN growth media (WGM: Opti-MEM supplemented with 0.5% FBS, 0.3% BSA, 100 μg/ml $CaCl_2$, 100 U/ml penicillin, and 100 μg/ml of streptomycin) for viral infections.

For an avian cell line, we chose to use a duck rather than a chicken cell line because ducks are natural hosts of influenza that (unlike chickens) possess RIG-I, a key innate-immune sensor of influenza (*Barber et al., 2010*).

For expansion of helper virus, we generated MDCK-SIAT1 cells expressing S009 PB2-E627K under control of a doxycycline-inducible promoter (MDCK-SIAT1-tet-S009-PB2-E627K) using a Sleeping Beauty transposon system (*Kowarz et al., 2015*). Briefly, MDCK-SIAT1 cells were transfected with pSBtet_RP_S009_PB2_E627K and pSB100X transposase vector using Lipofectamine 3000 (ThermoFisher Scientific, L3000015), and then subject to selection with 1 μg/ml puromycin. At three days post-transfection, we sorted for individual transfected cells expressing mCherry. All subsequent experiments were performed with a clonal expansion of a single transfected cell.

A549 cells were authenticated using the ATCC STR profiling service. CCL-141 cells were obtained from ATCC and used without extensive passaging. All cell lines tested negative for mycoplasma at the time they were expanded for either generating helper virus or passaging mutant plasmid libraries.

## Plasmids

Sequences for plasmids generated in this study are provided in *Supplementary file 1*.

Avian influenza polymerase plasmids: Original plasmids for PB2, PB1, PA, and NP genes from avian influenza strain A/Green-winged Teal/Ohio/175/1986 (S009) were gifts of Jeffrey Taubenberger (*Jagger et al., 2010*). For generating the mutant plasmid library, we cloned the S009 PB2 coding sequence into a pHW2000 vector (*Hoffmann et al., 2000*) from which we removed the CMV promoter (final plasmid pHW_noCMV_S009_PB2). The mutant library insert was cloned into the recipient vector pHW_noCMVnoTerm_BsmBI, which lacks the Pol I terminator (terminator sequence is part of the insert). The reason that we generated a pHW plasmid without a CMV promoter is that we were unable to maintain a stable bacterial clone of the S009 PB2 coding sequence on the pHH21 plasmid backbone – we observed frequent deletions in the coding sequence during plasmid propagation, suggesting that the insert on the pHH21 plasmid backbone is toxic to the bacterial host. For generating helper virus and virus for viral competitions, we cloned the S009 PB2, PB1, PA, and NP coding sequences into pHW2000 (pHW_S009_PB2, pHW_S009_PB1, pHW_S009_PA, pHW_S009_NP). In all cases we used non-coding viral-RNA termini from the respective A/WSN/1933 (H1N1) gene segment. For protein expression and the minigenome assay, we cloned the PB2, PB1, PA, and NP coding sequences from S009 into a protein-expression plasmid with a CMV promoter (HDM_S009_PB2, HDM_S009_PB1, HDM_S009_PA, HDM_S009_NP). All mutants of PB2 were made by site-directed mutagenesis on the appropriate plasmid backbone.

Helper virus plasmids: To generate a PB2 vRNA lacking a functional PB2 protein, we cloned GFP flanked by PB2 sequence into the pHH21 vector (*Neumann et al., 1999*) (pHH_PB2_S009_flank_99_eGFP_100). The flanking non-coding viral-RNA termini are from WSN PB2, and the coding sequences are from S009 PB2. The length of flanking sequences, 99 and 100 bases on the 5' and 3' end of the PB2 coding sequence respectively, are based on prior experiments analyzing how much terminal sequence is needed for effective genome packaging (*Liang et al., 2005*). We mutated out start codons 5' to the GFP start site in the mRNA sense. Our helper virus rescue also required the reverse genetics plasmids encoding HA, NA, M, and NS from WSN (pHW184_HA, pHW186_NA, pHW187_M, pHW188_NS) (*Hoffmann et al., 2000*). To generate a cell line with doxycycline-inducible expression of S009 PB2 for expansion of PB2-deficient helper virus, we cloned the S009 PB2-E627K coding sequence into the pSBtet vector (pSBtet_RP_S009_PB2_E627K) (*Kowarz et al., 2015*).

Minigenome assay: In addition to protein expression plasmids described above, we used a pHH-PB1-flank-eGFP reporter (*Bloom et al., 2010*), and pcDNA3.1-mCherry as a transfection control.

## Primers

All primer sequences used in this study are provided in *Supplementary file 2*. Note that this Excel file has several worksheets giving primers for different aspects of the experiments.

## PB2 codon mutant plasmid libraries

We generated all possible codon mutations of the entire PB2 coding sequence using the PCR-based strategy described in *Bloom (2014)* with the modifications described in *Dingens et al. (2017)*. Briefly, we designed mutagenic primers tiling across the entire coding region (https://github.com/jbloomlab/CodonTilingPrimers; *Bloom and Dingens, 2019*; copy archived at https://github.com/eli-fesciences-publications/CodonTilingPrimers). We performed 10 cycles of fragment PCR using the mutagenic primers and end primers flanking the vRNA, followed by 20 cycles of joining PCR using only end primers (*Supplementary file 2*: Mutagenesis worksheet). We generated three independent libraries starting from mutagenesis of independent bacterial clones. The PB2 variants were cloned into the BsmBI-digested vector pHW_noCMVnoTerm_BsmBI using NEBuilder HiFi DNA Assembly Master Mix (NEB, E2621S), and electroporated into ElectroMAX DH10B competent cells (Invitrogen, 18290015). We obtained 18–22 million transformants for each replicate library, from which we extracted plasmid by maxiprep. We randomly selected 48 clones for Sanger sequencing to evaluate the library mutation rate (https://github.com/jbloomlab/SangerMutantLibraryAnalysis; *Bloom et al., 2019*; copy archived at https://github.com/elifesciences-publications/SangerMutantLibraryAnalysis). (*Figure 1—figure supplement 2A–F*).

## Generation and passaging of mutant virus libraries

We generated mutant virus libraries using the helper-virus approach in *Doud and Bloom (2016)*, with modifications. We rescued reassortant virus using polymerase and nucleoprotein genes (PB2, PB1, PA, NP) from S009, and remaining genes (HA, NA, M, NS) from A/WSN/1933(H1N1) (WSN).

Helper virus: We plated a co-culture of $4 \times 10^5$ HEK293T and $0.5 \times 10^5$ MDCK-SIAT1-tet-S009-PB2-E627K in D10 media per well of a 6-well plate. At 18 hr after seeding cells, we added 1 µg/ml doxycycline to induce PB2 expression. One hour after adding doxycycline, we transfected each well with 250 ng each of pHH_PB2_S009_flank_99_eGFP_100, HDM_S009_PB2-E627K, pHW_S009_PB1, pHW_S009_PA, pHW_S009_NP, pHW184_HA, pHW186_NA, pHW187_M, and pHW188_NS using BioT (Bioland Scientific, B01-01). At four hours post-transfection, we replaced D10 media with WGM supplemented with 1 µg/ml doxycycline. We collected viral supernatant 52 hr post-transfection. To expand the helper virus, we seeded MDCK-SIAT1-tet-S009-PB2-E627K cells in D10 media 4 hr prior to infection at $4 \times 10^6$ cells per 15 cm dish. We then infected each dish with 40 µl of fresh viral supernatant, using WGM supplemented with 1 µg/ml doxycycline to induce PB2 expression. At 48 hr post-infection, we collected viral supernatant containing expanded helper virus and clarified the supernatant by centrifugation at 400x g for 4 min. We measured the infectious particle (IP)/µl titer of the helper virus by infecting HEK293T cells with a known volume of viral supernatant, and quantifying the number of GFP+ cells by flow cytometry 18 hr post-infection.

Mutant virus library rescue: For each mutant plasmid library, we seeded 36 wells of a 6-well dish with $1 \times 10^6$ HEK293T cells, and transfected each well 17 hr later with 375 ng each of HDM_S009_PB2-E627K, HDM_S009_PB1, HDM_S009_PA, HDM_S009_NP, and 500 ng of PB2 mutant plasmid library using BioT. For the wild-type control, we seeded six wells and used pHW_noCMV_S009_PB2 in place of the mutant plasmid library. At 6 hr post-transfection, we infected cells with helper virus at MOI of 1 IP/cell in WGM. At 2 hr post-infection, we replaced the inoculum with fresh WGM. At 20 hr post-infection, we harvested viral supernatants and clarified the supernatant by centrifugation at 400x g for 4 min. Supernatants were titered by $TCID_{50}$ on MDCK-SIAT1 cells. The titers for the three library replicates and wild-type control were 262, 68, 100, and 1467 $TCID_{50}$/µl respectively.

Passaging of mutant virus libraries: We aimed to passage $1 \times 10^6$ $TCID_{50}$ of each mutant virus library in A549 and CCL-141 cells at MOI of 0.01 $TCID_{50}$/cell as determined in MDCK-SIAT1 cells. Therefore, we aimed to have $1 \times 10^8$ total cells each at the time of infection. The day prior to each infection, we seeded between $8 \times 10^7$ to $1 \times 10^8$ A549 cells in D10 in $4 \times 5$ layered cell culture flasks (Corning, 353144), and $8 \times 10^7$ to $1 \times 10^8$ CCL-141 cells in E10 in $8 \times 5$ layered cell culture flasks. To estimate the total number of each cell type at the time of each infection, we plated an

equivalent density of cells in a T225 flask. Just prior to infection, we counted the number of cells in the T225 flask, and extrapolated the total number of cells to be infected. We calculated, for an MOI of 0.01, the number of $TCID_{50}$s to be passaged for each of the three library replicates and wild-type control in A549 cells to be $9.39 \times 10^5$, $1.02 \times 10^6$, $1.20 \times 10^6$, and $9.30 \times 10^5$ respectively, and in CCL-141 cells to be $8.46 \times 10^5$, $8.92 \times 10^5$, $9.54 \times 10^5$, and $8.58 \times 10^5$ respectively. We infected cells by removing D10 or E10 media, rinsing each flask with PBS, and then adding the calculated amount of virus diluted in WGM. At 3 hr post-infection, we replaced the inoculum with fresh WGM. The low MOI passage is expected to purge any non-replicative virus, including those containing the GFP segment. We confirmed this by our observation that there is limited spread of GFP expression over the course of infection. We harvested viral supernatant 48 hr post-infection, and clarified the supernatant by centrifugation at 400x g for 4 min.

## Barcoded subamplicon sequencing

We ultracentrifuged clarified viral supernatant at 27,000 rpm in a Beckman Coulter SW28 rotor, for 2 hr at 4°C. We resuspended the virus the residual media, then extracted RNA from 280 µl of concentrated virus using the Qiagen RNeasy Mini Kit (Qiagen, 74104). We titered the concentrated viral supernatant by $TCID_{50}$ on MDCK-SIAT1 cells to estimate the total $TCID_{50}$ from which we extracted RNA (ranged from $2.80 \times 10^5$ to $8.8 \times 10^6$ $TCID_{50}$); we expect the $TCID_{50}$ titer to be a lower-bound and underestimate of the total viral variants present in the supernatant, since it measures only infectious virus, whereas we would extract RNA from both infectious and non-infectious virions.

We used a barcoded-subamplicon deep sequencing strategy that reduces the sequencing error rate (*Doud and Bloom, 2016*), https://jbloomlab.github.io/dms_tools2/bcsubamp.html). Briefly, we reverse transcribed the full PB2 vRNA with Accuscript Reverse Transcriptase (Agilent, 200820) (primer S009-PB2-full-1F, *Supplementary file 2*: Barcoded subamplicon sequencing worksheet), and then PCR amplified the full PB2 vRNA (primers S009-PB2-full-1F and S009-PB2-full-8R, *Supplementary file 2*) using KOD Host Start Master Mix (EMD Millipore, 71842), making sure to have amplified from an estimated $1 \times 10^7$ cDNA molecules. During this amplification, we observed that the band corresponding to full-length PB2 was generally more intense than the smaller bands likely corresponding to the PB2-GFP gene segment, as well as PB2 deletions, suggesting that full-length PB2 was the most prevalent (*Figure 1—figure supplement 3*). We then PCR amplified the PB2 gene in eight subamplicons using primers containing a random barcode to uniquely identify each template cDNA molecule (*Supplementary file 2*). Approximately $7.5 \times 10^5$ uniquely barcoded molecules from each subamplicon library were then amplified by primers that add Illumina sequencing adaptors (*Supplementary file 2*). Finally, these libraries were deep sequenced on an Illumina HiSeq 2500 using $2 \times 250$ bp paired-end reads to a target 3.3x coverage per barcode.

## Analysis of deep mutational scanning data

Deep mutational scanning sequence data was analyzed using dms_tools2 (https://jbloomlab.github.io/dms_tools2, version 2.3.0). The GitHub repository https://github.com/jbloomlab/PB2-DMS (*Soh and Bloom, 2019*; copy archived at https://github.com/elifesciences-publications/PB2-DMS) contains Jupyter notebooks that perform all steps of the analyses and provide detailed step-by-step explanations and plots. The README file explains the organization of the notebooks and other files. HTML renderings of the notebooks are provided in *Supplementary file 3*. Processed results on preferences, differential selection, and mutation effect are provided in *Figure 2—source data 1*.

Rescaling of preferences: We rescaled our preferences to match the stringency of selection in nature. To do so, we first asked how well the preferences measured by the deep mutational scanning in either human or avian cells describes evolution of PB2 in both human and avian hosts. We used the preferences to generate an experimentally informed codon substitution model (ExpCM) (*Bloom, 2017*; *Hilton et al., 2017*), and asked if the ExpCMs described PB2's natural evolution better than a standard phylogenetic substitution model. The ExpCMs using amino-acid preferences vastly outperformed standard phylogenetic substitution models, suggesting that our experiments do capture some of the natural evolutionary constraint on PB2 (*Supplementary file 1*). The ExpCM stringency parameter had a value of 2.5, indicating that natural selection favors the same amino acids as our experiments, but with greater stringency (*Hilton et al., 2017*). We thus rescaled our

preferences to match the stringency of selection in nature using the ExpCM stringency parameter, and use these rescaled preferences for all subsequent analyses (*Figure 2—figure supplement 1*, *Supplementary file 1*, *Figure 2—source data 1*).

## Minigenome activity

Minigenome assays were performed in biological triplicate (starting from independent bacterial clones of each PB2 mutant) in both A549 and HEK293T cells. We seeded $2.5 \times 10^4$ A549 or HEK293T cells per well of a 96-well plate. Cells were transfected the next day with 10 ng each of HDM_S009_PB2 (for the respective mutant), HDM_S009_PB1, HDM_S009_PA, HDM_S009_NP, 30 ng of pHH-PB1-flank-eGFP reporter, and 30 ng of pcDNA-mCherry as transfection control, using Lipofectamine 3000 (A549) or BioT (HEK293T). At 22 hr post-transfection, cells were trypsinized and analyzed by flow cytometry. We report minigenome activity as the percent of mCherry-positive cells that are GFP-positive.

## Viral competition

Mutant virus: We generated mutant virus by reverse genetics using pHW_S009_PB2 (for the respective mutant), pHW_S009_PB1, pHW_S009_PA, pHW_S009_NP, pHW184_HA, pHW186_NA, pHW187_M, and pHW188_NS. We seeded $5 \times 10^5$ HEK293T cells per well of a 6-well plate, and transfected cells the next day with 250 ng each of the 8 pHW plasmids using BioT. At 2 hr post-transfection, we replaced media with WGM. At 40 hr post-infection, we collected viral supernatant. Viruses were titered by measuring the number of transcriptionally active particles as determined by flow cytometry for HA expression in infected A549 and CCL-141 cells. Briefly, we infected A549 and CCL-141 cells with a known volume of viral supernatant, and quantified the number of HA+ cells 9 hr post-infection. Cells were stained for HA using the H17-L19 antibody (*Gerhard et al., 1981*), which reacts with the WSN HA (*Doud et al., 2017*).

Competitions: Viral competition assays were performed in biological duplicate (starting from independent bacterial clones of each PB2 mutant). For each competition, A549 and CCL-141 cells were infected with a mixture of wild-type and PB2-mutant virus at a 1:1 ratio of transcriptionally active particles as measured for that cell type. For samples collected at 10 hr post infection, we infected a minimum of $3.78 \times 10^5$ cells at MOI of 0.1. For samples collected at 48 hr post infection, we infected a minimum of $7.62 \times 10^5$ cells at MOI of 0.01. At 2 hr post-infection, we replaced either D10 or E10 media with fresh WGM. At 10 hr post-infection, cells infected at MOI of 0.1 were lysed in buffer RLT and cellular RNA was extracted using the Qiagen RNeasy Mini Kit. At 48 hr post-infection, we collected viral supernatant from cells infected at MOI of 0.01, and extracted RNA using the QIAamp Viral RNA Mini Kit (Qiagen, 52904).

Sequencing to determine mutant frequency: We reverse transcribed the full length PB2 vRNA from the extracted RNA with SuperScript III (ThermoFisher Scientific, 18080051) (primer S009-PB2-full-1F, *Supplementary file 2*: Viral competition). For each PB2 mutant, we PCR amplified from the cDNA the region of PB2 centered around that mutated codon site (*Supplementary file 2*). This product was then subject to a second PCR using primers that add Illumina sequencing adaptors. Finally, these libraries were deep sequenced on an Illumina MiSeq using 50 bp single-end reads. Computational analyses to quantify mutant versus wild-type frequency are provided in *Supplementary file 3*, and at https://github.com/jbloomlab/PB2-DMS.

## H7N9 phylogenetic analysis

The phylogenetic tree was generated using Nextstrain's augur pipeline (*Hadfield et al., 2018*), and ancestral state reconstruction and adjustment of branch lengths according to sequence isolation date were performed with TreeTime (*Sagulenko et al., 2018*). Ancestral state reconstruction was only performed for nucleotide states, and was not used to infer ancestral host states. Instead, we inferred host-state transitions associated with each branch of the tree in a way that leveraged the prior knowledge that most H7N9 viruses circulate in avian hosts, and that most human infections arise from direct avian-to-human transmissions (*Su et al., 2017*). Specifically, for each node in the tree, starting from the root, we gathered all tips descending from that node. If that clade included only human sequences, and its parent node also included only human sequences, then the current clade falls within a monophyletic human clade, and the branch leading to it was labeled human-to-

human. If the current clade includes only human sequences but the parent node includes non-human sequences, then the branch leading to the clade was labeled avian-to-human. If the current clade includes both human and non-human sequences, then the branch leading to the clade was labeled avian-to-avian. Mutations were classified as human if they occurred on human-to-human and avian-to-human branches, and are classified as avian if they occurred on avian-to-avian branches. Note that since H7N9 human influenza typically results from avian-to-human transmissions, the branches we label as human-to-human in reality likely arise from avian-to-human transmissions, but which we cannot accurately reconstruct due to insufficient sampling of avian sequences. For this reason, human-to-human and avian-to-human branches were grouped together. Further details are provided in Jupyter notebook at https://github.com/jbloomlab/PB2-DMS, or in *Supplementary file 3*.

### Accessibility of mutations

We calculated the accessibility, or mean nucleotide substitutions required to access an amino-acid mutation, from avian influenza PB2 sequences collected from 2015 through 2018. The accessibility of codon *c* to amino-acid *a* by single-nucleotide mutations is defined as the minimum number of nucleotide mutations needed to generate any codon for that amino-acid. For a collection of sequences, we calculate the accessibility as the weighted average of the accessibilities of all codons observed at that site in the collection of sequences. Accessibility was calculated using code as documented here (https://jbloomlab.github.io/dms_tools2/dms_tools2.utils.html?highlight=accessibility#dms_tools2.utils.codonEvolAccessibility), in Jupyter notebook at https://github.com/jbloomlab/PB2-DMS, and in *Supplementary file 3*.

### Quantification and statistical analysis

Quantification and statistical analysis was performed in Python and a complete description is available in main text, methods, associated figure legends, and computational Jupyter notebooks.

### Data and software availability

Deep sequencing data have been deposited in the NCBI Sequence Read Archive under BioProject accession number PRJNA511556.

The GitHub repository https://github.com/jbloomlab/PB2-DMS contains Jupyter notebooks that perform all steps of computational analyses and provide detailed step-by-step explanations and plots. The README file explains the organization of the notebooks and other files. HTML renderings of the notebooks are provided in *Supplementary file 3*.

## Acknowledgements

We thank William Fowler for help with molecular cloning, Juhye Lee, Adam Dingens, and John Huddleston for advice and computer code, and Harmit Malik, Katherine Xue, Allison Greaney, and Tyler Starr for providing comments on this manuscript. We thank Jeffrey Taubenberger for providing plasmids for A/Green-winged Teal/Ohio/175/1986 (S009), and Andrew Mehle and Steven Baker for advice about growing virus containing S009 polymerase and NP genes. YQSS was supported by the Mahan Postdoctoral Fellowship (Computational Biology Program of Fred Hutchinson Cancer Research Center) and the Damon Runyon Postdoctoral Fellowship. This work was supported by NIH grant R01 AI127893 from the NIAID to TB and JDB, and NIH grant R35 GM119774-01 from the NIGMS to TB. TB is a Pew Biomedical Scholar. JDB is an Investigator of the Howard Hughes Medical Institute.

## Additional information

### Funding

| Funder | Grant reference number | Author |
|---|---|---|
| National Institute of Allergy and Infectious Diseases | R01 AI127893 | Trevor Bedford Jesse D Bloom |
| National Institute of General Medical Sciences | R35 GM119774-01 | Trevor Bedford |

| Damon Runyon Cancer Research Foundation | DRG-2271-16 | Y Q Shirleen Soh |
| Pew Charitable Trusts | | Trevor Bedford |
| Howard Hughes Medical Institute | | Jesse D Bloom |
| Mahan Postdoctoral Fellowship | | Y Q Shirleen Soh |

The funders had no role in study design, data collection and interpretation, or the decision to submit the work for publication.

### Author contributions

YQ Shirleen Soh, Conceptualization, Software, Formal analysis, Validation, Investigation, Visualization, Methodology, Writing—original draft, Writing—review and editing; Louise H Moncla, Data curation, Software, Formal analysis, Methodology, Writing—review and editing; Rachel Eguia, Validation, Investigation, Writing—review and editing; Trevor Bedford, Data curation, Supervision, Funding acquisition, Methodology, Writing—review and editing; Jesse D Bloom, Conceptualization, Software, Supervision, Funding acquisition, Methodology, Writing—review and editing

### Author ORCIDs

YQ Shirleen Soh https://orcid.org/0000-0002-7422-7414
Trevor Bedford http://orcid.org/0000-0002-4039-5794
Jesse D Bloom http://orcid.org/0000-0003-1267-3408

### Decision letter and Author response

Decision letter https://doi.org/10.7554/eLife.45079.036
Author response https://doi.org/10.7554/eLife.45079.037

## Additional files

### Supplementary files

• Supplementary file 1. Plasmid sequences.
DOI: https://doi.org/10.7554/eLife.45079.028

• Supplementary file 2. Primer sequences.
DOI: https://doi.org/10.7554/eLife.45079.029

• Supplementary file 3. Jupyter notebooks documenting computational analyses.
DOI: https://doi.org/10.7554/eLife.45079.030

• Supplementary file 4. Comparison of ExpCM to standard phylogenetic substitution models.
DOI: https://doi.org/10.7554/eLife.45079.031

• Transparent reporting form
DOI: https://doi.org/10.7554/eLife.45079.032

### Data availability

Deep sequencing data have been deposited in the NCBI Sequence Read Archive under BioProject accession number PRJNA511556. All data generated or analyzed during this study are included in the manuscript and supporting files. Source data files have been provided for Figures 2, 3, 4, 6, and 7. The GitHub repository https://github.com/jbloomlab/PB2-DMS contains Jupyter notebooks that perform all steps of computational analyses and provide detailed step-by-step explanations and plots.

The following dataset was generated:

| Author(s) | Year | Dataset title | Dataset URL | Database and Identifier |
| --- | --- | --- | --- | --- |
| Soh YQS | 2019 | Deep mutational scanning of avian influenza PB2 to identify host- | https://www.ncbi.nlm.nih.gov/bioproject/? | NCBI BioProject, PRJNA511556 |

adaptive mutations                    term=PRJNA511556

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
