## [Decision Letter]

Thank you for submitting your article "Comprehensive mapping of avian influenza polymerase adaptation to the human host" for consideration by *eLife*. Your article has been reviewed by three peer reviewers, and the evaluation has been overseen by a Reviewing Editor and Neil Ferguson as the Senior Editor. The following individuals involved in review of your submission have agreed to reveal their identity: Ervin Fodor (Reviewer #1); Benjamin R. tenOever (Reviewer #3).

The reviewers have discussed the reviews with one another and the Reviewing Editor has drafted this decision to help you prepare a revised submission.

Summary:

Soh et al. present an impressive amount of work and data analyses to comprehensively map residues on the PB2 polymerase subunit of influenza A virus that confer species adaptation. The results of this work provide a map for the most likely host adaptive mutations that might enable an avian influenza A virus strain to replicate efficiency in mammals. Their screens not only identified known PB2 variants involved in the transition from avian to mammalian adaptation, but they also identified a number of novel residues that confer comparable properties. Interestingly, they note that many of the residues implicated in mammalian adaptation fall adjacent to known sites of host interactions and are on the exposed surface of PB2. In addition, they found that those residues that increase transcriptional activity but have not been found in nature generally demand numerous nucleotide substitutions thus making their selection more difficult. Overall, this work is based on a beautiful combination of deep sequencing, functional experiments, correlation to known structural information and evolutionary analyses, and the results may provide an important resource to those trying to predict what strains to monitor from general viral surveillance efforts. Although the underlying mechanisms governing preference for certain amino acids at certain positions remain unexplored, the study is an important addition to the body of literature on the host adaptive changes of the influenza virus RNA polymerase.

Essential revisions:

1) Figure 4. Please add statistical tests on the data in panels A, B, and C. For panel C, provide a more detailed description to help the reader to understand exactly what is plotted on the y-axis and how these data correspond with the two different MOIs used and total and viral RNA analyzed at 10 and 48 hpi (Subsection “Viral competition”, second paragraph).

2) Subsection “Human-adaptive mutations identified in deep mutational scanning improve polymerase activity and viral growth in human cells”, fifth paragraph. Of the mutations that have little to no effect on polymerase activity, only R355G has a convincing effect in the viral competition assays. I292T, S523D, and to a lesser degree R355K do not deviate significantly from the starting ratio of wild type/mutant, especially when considering the E627E negative control. Hence the data do not support the conclusion that these mutations enhance viral growth.

3) Subsection “Human-adaptive mutations cluster in regions of PB2 that are potentially important for host adaptation”, first paragraph, Figure 5 and Figure 5—figure supplement 1. Top human-adaptive mutations appear to cluster in regions that play a role in host adaptation. It is not clear whether these are all surface exposed; were there any mutations affecting internal PB2 residues?

4) Figure 6B. PB2 E627K is the most frequent adaptive mutation acquired by H7N9 influenza virus during avian-to-human transmission, but has a much lower differential selection than other less-frequent E627 mutations. Please address this apparent inconsistency.

5) Saturating mutagenesis has already been performed at the two major sites of adaption, PB2 627 and 701 (Chin et al., 2014, and Chin et al., 2017, respectively). Differential selection for the PB2 pool in these prior experiments also found some of the same "winners" reported here. In addition, PB2 S627 has also been reported previously. This variant is naturally encoded by bat influenza viruses and supports high-level polymerase activity in human cells (Tong et al., 2012; Tong et al., 2013) Please reference and incorporate these points into your analysis.

6) PB2 mutant viruses were rescued using a helper virus that encodes GFP in place of PB2. How is the PB2-like reporter gene eliminated from the mutant virus population, or does it persist? If it persists, how does the presence of what is effectively semi-infectious virus impact replication?

7) The Discussion is very perfunctory and primarily repeats points already raised in the Results section. A synthesis of the current findings with the existing literature would help place these findings in context.

8) Figure 2C-D: the authors don't comment on why the entropy in natural avian sequences is so much lower on average than in human sequences. As a non-expert this is what I noticed first about these panels. Perhaps this could be explained better in the text.

9) Their minigenome assay (Figure 4A-B) is technically only a reporter assay for transcription and not necessarily replication, which they don't mention.

10) Figure 4C, subsection “Human-adaptive mutations identified in deep mutational scanning improve polymerase activity and viral growth in human cells”, fourth paragraph: viral competition assay. the authors say they infect both viruses at a 1:1 ratio, but how do they know it is actually a true 1:1 ratio? The Materials and methods state this is based on the IP/ul titer. This reviewer has no idea what that means as this is certainly not a conventional way of measuring virus. Moreover, how do the authors account for DIs etc.?

11) Figure 5: I am not sure the authors have accounted for all of the published PB2 adaptations. A quick perusal on Pubmed identifies A44S, T105V, E158G, A199S, D253N, D256G, T271A, R368K, L475M, R493K, K526R, T559I, D567N, T569A, A588I, G590S, Q591R, V598T, V613T, A661T, A674T, G682S, A684S, and S714I as residues that have been implicated in avian-to-mammalian adaptation. Are the authors certain they are working off a comprehensive list of known adaptation residues? It seems incomplete.

---

## [Author Response]

Essential revisions:1) Figure 4. Please add statistical tests on the data in panels A, B, and C. For panel C, provide a more detailed description to help the reader to understand exactly what is plotted on the y-axis and how these data correspond with the two different MOIs used and total and viral RNA analyzed at 10 and 48 hpi (Subsection “Viral competition”, second paragraph).

We have now performed t-tests on panels A and B. We found that all mutations, except the ones we had previously noted as having little to no effect on polymerase activity, show significant effects (p < 0.05). We have indicated significant differences with an asterisk (*) on Figure 4A and B.

Experiments for Figure 4C were performed in biological duplicate, due to limitations on the number and scale of experiments. The duplicates do not provide enough statistical power to detect significant differences. However, the effects of mutations are evident by visual inspection of the two replicates, each of which were used in two independent infections and sampled at two different timepoints.

For Panel C, we have clarified in the figure itself and the text that the data plotted on the y-axis is the ratio of mutant:WT virus, in A549 over CCL-141 cells. We have also removed from the figure the connecting lines between the 10 hpi and 48 hpi datapoints, so as to clarify that the data at 10 hpi and 48 hpi are from separate infections with two different MOIs.

2) Subsection “Human-adaptive mutations identified in deep mutational scanning improve polymerase activity and viral growth in human cells”, fifth paragraph. Of the mutations that have little to no effect on polymerase activity, only R355G has a convincing effect in the viral competition assays. I292T, S523D, and to a lesser degree R355K do not deviate significantly from the starting ratio of wild type/mutant, especially when considering the E627E negative control. Hence the data do not support the conclusion that these mutations enhance viral growth.

We have now revised our description of the data to state that only R355G, and to a lesser degree R355K, have strong effects on viral growth among mutations that do not affect polymerase activity.

3) Subsection “Human-adaptive mutations cluster in regions of PB2 that are potentially important for host adaptation”, first paragraph, Figure 5 and Figure 5—figure supplement 1. Top human-adaptive mutations appear to cluster in regions that play a role in host adaptation. It is not clear whether these are all surface exposed; were there any mutations affecting internal PB2 residues?

Most of the sites of adaptive mutations are surface exposed in at least one of the two conformations of the polymerase. We have revised the text to clarify which sites are and are not surfaced exposed. We have also added an analysis of the relative solvent accessibility of these sites to Figure 5—figure supplement 1. Only mutations at sites 163 and 182 affect internal PB2 residues and are therefore not exposed in any known conformation of PB2. In addition, site 156, although on the surface of the PB2 protein, faces the internal core of the polymerase in both conformations examined, and is thus likely also inaccessible.

4) Figure 6B. PB2 E627K is the most frequent adaptive mutation acquired by H7N9 influenza virus during avian-to-human transmission, but has a much lower differential selection than other less-frequent E627 mutations. Please address this apparent inconsistency.

We have added to the Discussion some possible reasons for inconsistencies between our deep mutational scanning measurements and the frequencies of mutations observed in nature. In particular, we address the reviewer’s question of why the E627K mutation is most frequently observed in nature, though it has lower differential selection than other site 627 mutations. First, many mutations with higher differential selection than mutations we observe in nature are inaccessible from current avian influenza sequences by single-nucleotide mutations. This makes them much less likely to arise. Amongst mutations that are accessible by single-nucleotide mutations, such as E627K and E627V, the former is more frequently observed despite having lower differential selection. One possible explanation is that the rate of single-nucleotide mutations can differ depending on whether it is a transition or transversion. For example, the E627K mutation requires a transition from base G to A, whereas E627V requires a transversion from base A to T. Transitions occur at a higher rate than transversions. Specifically, previous measurements of influenza mutation rates estimates that G to A transitions occur at ten times the rate of A to T transversions. Finally, it is possible that other factors which are not captured by our in vitro model preferentially select for E627K.

5) Saturating mutagenesis has already been performed at the two major sites of adaption, PB2 627 and 701 (Chin et al., 2014, and Chin et al., 2017, respectively). Differential selection for the PB2 pool in these prior experiments also found some of the same "winners" reported here. In addition, PB2 S627 has also been reported previously. This variant is naturally encoded by bat influenza viruses and supports high-level polymerase activity in human cells (Tong et al., 2012; Tong et al., 2013) Please reference and incorporate these points into your analysis.

We have revised the text to incorporate these references.

6) PB2 mutant viruses were rescued using a helper virus that encodes GFP in place of PB2. How is the PB2-like reporter gene eliminated from the mutant virus population, or does it persist? If it persists, how does the presence of what is effectively semi-infectious virus impact replication?

The mutant virus library (still containing PB2-GFP gene segments) was passaged at very low MOI after rescue so as to purge any non-replicative virus, including those containing the GFP segment. We confirmed this by our observation that there is limited spread of GFP expression over the course of infection. We have now explained this in the Materials and methods section. During amplification of the PB2 gene segment for sequencing library preparation, we also observed that the band corresponding to full-length PB2 was generally more intense than the smaller bands likely corresponding to the PB2-GFP gene segment, as well as PB2 deletions, suggesting that full-length PB2 was the most prevalent. We suspect that the relative amounts of full-length PB2 could be even higher than indicated by the gel, as the product underwent 22 cycles of PCR, which is known to preferentially amplify shorter templates (such as PB2-GFP and PB2-deletions). We now provide this gel image as Figure 1—figure supplement 3.

7) The Discussion is very perfunctory and primarily repeats points already raised in the Results section. A synthesis of the current findings with the existing literature would help place these findings in context.

We have now expanded our discussion of the current findings in context of existing literature. We explain how our complete map of adaptive mutations and amino acid preferences provides context for previously identified mutations that affect host adaptation and PB2 function. We explain that our analyses of new adaptive mutations in combination with previously identified mutations and published crystal structures suggest that host adaptation may involve mutations at sites at the periphery of core interactions. Finally, we also expand our discussion of why the magnitude of differential selection as quantified in our experiments do not always correspond to observed frequencies of mutations in nature that have been described in the literature, as explained in our response to comment 4 above.

8) Figure 2C-D: the authors don't comment on why the entropy in natural avian sequences is so much lower on average than in human sequences. As a non-expert this is what I noticed first about these panels. Perhaps this could be explained better in the text.

We have now included an explanation in the text describing Figure 2. Entropy, a measure of sequence variation, is generally low in natural avian sequences, probably because influenza A virus is highly adapted to avian hosts so there is little pressure for additional adaptation. Entropy is generally higher in natural human sequences, likely because of increased genetic diversity generated as a result of directional selection to adapt to the human host (dos Reis et al., 2010), and diversifying selection to escape immune selection on PB2-derived T-cell epitopes (e.g. Assaarsson et al., 2008).

9) Their minigenome assay (Figure 4A-B) is technically only a reporter assay for transcription and not necessarily replication, which they don't mention.

We have updated the text to emphasize that the minigenome assay only quantifies the transcription of mRNA by the viral polymerase.

10) Figure 4C, subsection “Human-adaptive mutations identified in deep mutational scanning improve polymerase activity and viral growth in human cells”, fourth paragraph: viral competition assay. the authors say they infect both viruses at a 1:1 ratio, but how do they know it is actually a true 1:1 ratio? The Materials and methods state this is based on the IP/ul titer. This reviewer has no idea what that means as this is certainly not a conventional way of measuring virus. Moreover, how do the authors account for DIs etc.?

We have now clarified that the IP/ul titer reflects transcriptionally active particles as determined by flow cytometry for HA expression in infected cells. We also clarify that cells were infected with a 1:1 ratio of transcriptionally active particles, to the best of our measurements. Our measurements of transcriptionally active particles includes potential DIs which are competent to infect and express HA, but are not necessarily fully infectious.

More importantly, as we clarify also in response to Comment 1, the data plotted in Figure 4C is the ratio of mutant:WT virus, in A549 over CCL-141 cells. So even if the ratio of mutant:WT infectious virus is not exactly 1:1, it is corrected for since we compare the ratio in two cell types. Therefore, the quantity on the y-axis is only sensitive to *relative* differences of the wildtype and mutant variants in the two cell types, not their exact ratio in any single cell type.

11) Figure 5: I am not sure the authors have accounted for all of the published PB2 adaptations. A quick perusal on Pubmed identifies A44S, T105V, E158G, A199S, D253N, D256G, T271A, R368K, L475M, R493K, K526R, T559I, D567N, T569A, A588I, G590S, Q591R, V598T, V613T, A661T, A674T, G682S, A684S, and S714I as residues that have been implicated in avian-to-mammalian adaptation. Are the authors certain they are working off a comprehensive list of known adaptation residues? It seems incomplete.

We have double-checked and confirmed that most of the mutations listed above by the reviewer were included in our analyses. We did miss a few, which we have now included.

We now clarify that in Figure 5 we plot in blue adaptive mutations for which there is prior experimental support (total of 25 mutations). We also clarify that we have labeled only a select few of them as landmarks to refer to in the text. (There is not enough space to label all of them.)

We provide a full list of published PB2 mutations, both with and without prior experimental support, in Figure 3—source data 1. Our analyses use only those mutations which have been experimentally validated to improve polymerase activity or viral growth in a human or mammalian host. This is because some computationally identified host adaptive mutations have failed to experimentally validate (as reported in previous published studies). We also required that each mutation showed an adaptive effect on its own, rather than in combination with another adaptive mutation, so as to make a fair comparison to our experiments which aim to identify effects of single mutations.

Of the mutations listed above, the following were already included in our analyses as they have experimental support: T158G, A199S, D253N, D256G, T271A, K526R, A588I, Q591K, A684S.

Of the mutations listed above, the following were listed in Figure 3—source data 1, but were not used in our analyses as they are not yet supported by experimental evidence. We have repeated literature searches to confirm that there has been no new evidence since: A44S (did not improve polymerase activity when tested in Cauldwell et al., 2013), T105V, R368K (enhances pathogenicity only in combination with E627K), L475M, D567N, G590S (not yet tested separately from Q591R), Q591R (not yet tested separately from G590S), V613T, A661T, A674T (did not improve polymerase activity or viral replication alone, Lee et al., 2017), and G682S.

We did miss the following mutations: R493K, T559I, T569A, V598T, S714I. Of these, we found experimental support for only T569A. We found that S714I had been experimentally tested but showed rather modest effect (Czudai-Matwich et al., 2014). Another mutation at this same site, S714R was already included in our analyses as having experimental support. We have added these missed mutations to Figure 3—source data 1. We have also updated Figure 5 (as well as Figure 3) to include the one missed mutation with experimental support (T569A).